# Estimation and worldwide monitoring of the effective reproductive number of SARS-CoV-2

Jana S Huisman[1,2,3]*[†], Jérémie Scire[2,3†], Daniel C Angst[1], Jinzhou Li[4], Richard A Neher[2,5], Marloes H Maathuis[4], Sebastian Bonhoeffer[1‡], Tanja Stadler[2,3]*[‡]

[1]Department of Environmental Systems Science, ETH Zurich, Swiss Federal Institute of Technology, Zurich, Switzerland; [2]Swiss Institute of Bioinformatics, Lausanne, Switzerland; [3]Department of Biosystems Science and Engineering, ETH Zurich, Swiss Federal Institute of Technology, Basel, Switzerland; [4]Department of Mathematics, ETH Zurich, Swiss Federal Institute of Technology, Zurich, Switzerland; [5]Biozentrum, University of Basel, Basel, Switzerland

*For correspondence:
jana.huisman@env.ethz.ch (JSH);
tanja.stadler@bsse.ethz.ch (TS)

[†]These authors contributed equally to this work
[‡]These authors also contributed equally to this work

**Abstract** The effective reproductive number $R_e$ is a key indicator of the growth of an epidemic. Since the start of the SARS-CoV-2 pandemic, many methods and online dashboards have sprung up to monitor this number through time. However, these methods are not always thoroughly tested, correctly placed in time, or are overly confident during high incidence periods. Here, we present a method for timely estimation of $R_e$, applied to COVID-19 epidemic data from 170 countries. We thoroughly evaluate the method on simulated data, and present an intuitive web interface for interactive data exploration. We show that, in early 2020, in the majority of countries the estimated $R_e$ dropped below 1 only after the introduction of major non-pharmaceutical interventions. For Europe the implementation of non-pharmaceutical interventions was broadly associated with reductions in the estimated $R_e$. Globally though, relaxing non-pharmaceutical interventions had more varied effects on subsequent $R_e$ estimates. Our framework is useful to inform governments and the general public on the status of epidemics in their country, and is used as the official source of $R_e$ estimates for SARS-CoV-2 in Switzerland. It further allows detailed comparison between countries and in relation to covariates such as implemented public health policies, mobility, behaviour, or weather data.

## Editor's evaluation

Understanding the trajectory of epidemic growth and predicting it in real-time is an important goal of epidemiological modelling. This work aggregates data from 170 countries in an effort to better understand how the effective reproduction number of SARS-CoV-2 spread evolved over time and across the world.

## Introduction

During an infectious-disease outbreak, such as the SARS-CoV-2 pandemic, accurate monitoring of the epidemic situation is critical to the decision-making process of governments and public health authorities. The magnitude of an epidemic, as well as its spatial and temporal infection dynamics determine the exposure risk posed to citizens in the near and long-term future, the pressure on critical infrastructure like hospitals, and the overall burden of disease to society.

**eLife digest** Over the past two and a half years, countries around the globe have struggled to control the transmission of the SARS-CoV-2 virus within their borders. To manage the situation, it is important to have an accurate picture of how fast the virus is spreading. This can be achieved by calculating the effective reproductive number (Re), which describes how many people, on average, someone with COVID-19 is likely to infect. If the Re is greater than one, the virus is infecting increasingly more people, but if it is smaller than one, the number of cases is declining.

Scientists use various strategies to estimate the Re, which each have their own strengths and weaknesses. One of the main difficulties is that infections are typically recorded only when people test positive for COVID-19, are hospitalized with the virus, or die. This means that the data provides a delayed representation of when infections are happening. Furthermore, changes in these records occur later than measures that change the infection dynamics. As a result, researchers need to take these delays into account when estimating Re.

Here, Huisman, Scire et al. have developed a new method for estimating the Re based on available data records, statistically taking into account the above-mentioned delays. An online dashboard with daily updates was then created so that policy makers and the population could monitor the values over time.

For over two years, Huisman, Scire et al. have been applying their tool and dashboard to COVID-19 data from 170 countries. They found that public health interventions, such as mask requirements and lockdowns, did help reduce the Re in Europe. But the effects were not uniform across the globe, likely because of variations in how restrictions were implemented and followed during the pandemic. In early 2020, the Re only dropped below one after countries put lockdowns or other severe measures in place.

The Re values added to the dashboard over the last two years have been used pro-actively to inform public health policies in Switzerland and to monitor the spread of SARS-CoV-2 in South Africa. The team has also recently released programming software based on this method that can be used to track future disease outbreaks, and extended the method to estimate the Re using SARS-CoV-2 levels in wastewater.

---

The effective reproductive number $R_e$ is a key indicator to describe how efficiently a pathogen spreads in a given population at a given time (*Anderson and May, 1991*; *Cauchemez et al., 2006*; *Wallinga and Lipsitch, 2007*). It quantifies the average number of secondary infections caused by a primary infected individual. It also has a natural threshold value of 1, below which the epidemic reduces in size *Anderson and May, 1991*; *Nishiura and Chowell, 2009*. $R_e$ typically changes during the course of an epidemic as a result of the depletion of susceptible individuals, changed contact behaviour, seasonality of the pathogen, or the effect of pharmaceutical and non-pharmaceutical interventions (NPIs) (*Anderson and May, 1991*; *Delamater et al., 2019*; *Scire et al., 2020*; *Ali et al., 2020*; *Flaxman et al., 2020*).

Different methods have been developed to estimate $R_e$. They broadly fall into two categories: those based on compartmental models, (e.g. *Delamater et al., 2019*; *Kucharski et al., 2020*; *Zhou et al., 2020*), and those that infer the number of secondary infections per infected individual directly, based on a time series of infection incidence, (e.g. *Wallinga and Teunis, 2004*; *Cori et al., 2013*). We focus on the latter class of methods as they rely on few, simple assumptions, are less prone to model misspecifications, and are well-suited for ongoing monitoring of the epidemic (*Gostic et al., 2020*). In particular, we consider the EpiEstim method of *Cori et al., 2013*.

The infection incidence based methods face the difficulty that infection events cannot be observed directly (*Gostic et al., 2020*). These events can only be surmised with a certain time lag, e.g. when individuals show symptoms and are tested, via contact tracing, or via periodic testing of a cohort of individuals (*Nishiura and Chowell, 2009*). To use these methods, one must thus employ a proxy for infection events (e.g. the observed incidence of confirmed cases, hospitalisations, or deaths). This proxy is either used directly in lieu of the infection incidence, or it is used as an indirect observation to infer past infections (*Gostic et al., 2020*). It is important to relate $R_e$ estimates to the timing of infection events because this allows multiple proxies of infection events, with differing delays, to be

used separately to monitor the same epidemic (*Scire et al., 2020*). In addition, any factors that may affect transmission dynamics will do so at the time infections occurred. If $R_e$ is placed properly on this timescale, it can be compared directly to external covariates like weather and interventions (*Flaxman et al., 2020*; *Soltesz et al., 2020*). However, depending on the method used to infer the timing of infections from the observed incidence time series, one can also introduce biases such as smoothing sudden changes in $R_e$ (*Gostic et al., 2020*; *Goldstein et al., 2009*; *Petermann and Wyler, 2020*).

Several methods, software packages, and online dashboards have been developed to monitor developments in $R_e$ during the SARS-CoV-2 pandemic (e.g. *Abbott et al., 2020b*; *Systrom et al., 2020*; *Tebé et al., 2020*; *Scott et al., 2020*; *Hamouda, 2020*; *Richter et al., 2020*). A pipeline for the continuous estimation of $R_e$ using infection incidence based methods should include four critical steps: (i) gathering and curating observable proxy data of infection incidence, (ii) reconstruction of the unobserved infection events, (iii) $R_e$ estimation, and (iv) communication of the results, including uncertainty and potential biases. These four axes also define the differences between existing methods. The first step dictates e.g. the geographical scope of the $R_e$ estimates reported. During the SARS-CoV-2 epidemic, many local public health authorities have made case data publicly available. Depending on the data sources used, estimated $R_e$ values span from the scale of a city, region, country, or the entire globe (*Systrom et al., 2020*; *Pan et al., 2020*; *Robert Koch-Institut, 2020*). The second step, i.e. going from a noisy time series of indirect observations to an infection incidence time series, is technically challenging. Biases can be introduced easily, and accurately assessing the uncertainty around the inferred infection incidence is a challenge in itself (*Gostic et al., 2020*). For the third step, i.e. to estimate $R_e$ from a timeline of infection events, there are ready-to-use software packages (*Cori et al., 2013*; *Obadia et al., 2012*), which produce $R_e$ estimates along with an estimate of the uncertainty resulting from this step. Finally, the communication of results to the general public and decision makers is essential, but often overlooked. We present a pipeline, together with an online dashboard, for timely monitoring of $R_e$. We use publicly available data gathered by different public health authorities. Wherever possible, we show results obtained from different types of case reports (confirmed cases, hospitalisations or deaths). This allows comparison across observation types and to balance the biases inherent in the different types. Results are updated daily, and can be found on https://ibz-shiny.ethz.ch/covid-19-re-international/. The results of this method have been used directly in public health policy making in Switzerland for the past 2 years, and were also communicated by the Federal Office of Public Health on https://www.covid19.admin.ch/en/overview during that time. Through continuous engagement with the public, scientific experts, and thorough evaluation on simulated scenarios, we have created a robust and transparent method of enduring relevance for the current and future epidemics.

Because $R_e$ estimates reflect changes in virus transmission dynamics, they can be used to assess the impact of public health interventions. Prior work on the relative impact of specific non-pharmaceutical interventions on $R_e$ has shown conflicting results (*Flaxman et al., 2020*; *Esra et al., 2020*; *Banholzer et al., 2021*; *Soltesz et al., 2020*; *Haug et al., 2020*; *Kohanovski et al., 2022*). These differences can be attributed mostly to different model formulations (*Soltesz et al., 2020*; *Sharma et al., 2020*; *Banholzer et al., 2022*), including differing assumptions on the independence of NPIs (*Sharma et al., 2020*), differing timescales over which the effect of the NPI was analysed (*Flaxman et al., 2020*; *Haug et al., 2020*), whether the time point of the NPI was assumed fixed or allowed to vary (*Kohanovski et al., 2022*), and differing geographical scope. There is a need to address whether the strength of measures and the speed of their implementation resulted in a larger and faster decrease of $R_e$, and specifically whether highly restrictive lockdowns were necessary to achieve $R_e < 1$. Further, it remains unclear how the impact of interventions differed across time and geographical regions. We add to this debate by using our $R_e$ estimates across geographical regions and timescales that include the lifting of many NPIs. While we cannot determine causal relationships, we use our method to assess likely associations.

## Results

## A pipeline to estimate the effective reproductive number of SARS-CoV-2

We have developed a pipeline to estimate the time-varying effective reproductive number of SARS-CoV-2 from observed COVID-19 case incidence time series (see Materials and Methods). The objective was to achieve stable estimates for multiple types of data, and with an adequate representation of uncertainty. At the core, we use the EpiEstim method (*Cori et al., 2013*) to estimate $R_e$ from a time series of infection incidence. To infer the infection incidence from a time series of (noisy) observations, we extended the deconvolution method by Goldstein et al. to deal with partially observed data and time-varying delay distributions (*Gostic et al., 2020*; *Goldstein et al., 2009*). We smooth the data prior to deconvolution to reduce numerical artefacts resulting from their weekly patterns and overall noisy nature. We compute point-wise 95% confidence intervals for the true $R_e$ values, using the union of a block bootstrap method, designed to account for variation in the case observations, and the credible intervals from EpiEstim. As observed incidence data we use COVID-19 confirmed case data, hospital admissions, and deaths (with type specific delay distributions, see Materials and Methods). We publish separate $R_e$ estimates for each of these types of incidence data. The most recent $R_e$ estimate lies further in the past than the most recent observed incidence data due to the delay between infection and case observation.

## Evaluation on simulated data

To evaluate our method, we used simulations of several epidemic scenarios (see Materials and Methods for more details). For each scenario, we specified an $R_e$ time series. The specified $R_e$ trajectories were parametrised in a piecewise linear fashion. To mimic the course of the COVID-19 outbreaks observed in many European countries in 2020 (*Lemaitre et al., 2020*), we started with $R_e$ values around 3, then dropped to a value below 1 (the 'initial decrease'), stayed around 1 in summer and slightly above 1 (the 'second wave') in autumn (*Figure 1*). From each specified $R_e$ trajectory, we stochastically simulated 100 time series of infections and their resulting case observations. To account for reporting effects and to better mimic observed COVID-19 case data from around the world, we added additional autocorrelated noise to the case observations.

We then used our method to infer the infection incidence and $R_e$ from each simulated time series of case observations, and compared these to the true underlying $R_e$ values (*Figure 1*). The results show that the method accurately estimates the effective reproductive number (*Figure 1*; metrics described in Materials and Methods). Across most time points, the 95% confidence interval includes the true $R_e$ value (coverage; *Figure 1B*). The low root mean squared error (RMSE) indicates that our point estimates closely track the true $R_e$ value (*Figure 1C*). Importantly, we correctly infer whether $R_e$ is significantly above or below 1 in this scenario: we never infer that $R_e$ is significantly above 1 when the true value is below 1, and only for two time points the estimates are significantly below 1 for some simulations when the true value is a little above 1 (*Figure 1D*). Due to the smoothing step prior to deconvolution, we slightly misestimate $R_e$ during steep changes (see Appendix 2, and *Appendix 3—figure 1* for more scenarios). However, the inclusion of smoothing greatly improves the performance across scenarios with different types of observation noise (*Appendix 3—figures 2 and 3*). For a wide range of infection incidences, the 95% confidence interval is informative and covers the true value of $R_e$ (*Appendix 3—figures 4 and 5*). Our block-bootstrapping method greatly improves coverage compared to the out-of-the-box EpiEstim method. Our method also outperforms the common approach of using a fixed delay to infer the infection incidence time series (*Appendix 3—figure 6*).

We further tested the impact of misspecifying the delay between infections and observations. Misspecifying the mean of the delay distribution between infection and case observation by up to 2 days does not have a strong effect on the $R_e$ estimates, whereas larger misspecifications by 5 or 10 days lead to a more substantial decrease in coverage (*Appendix 3—figure 7*). Correspondingly, allowing for time-varying delay distributions has a pronounced effect on the estimated $R_e$ only when changes in the mean of the delay distribution are large (*Appendix 3—figure 8*). The impact of other model misspecifications, e.g. of the generation time interval, have been investigated by *Gostic et al., 2020* for this class of methods.

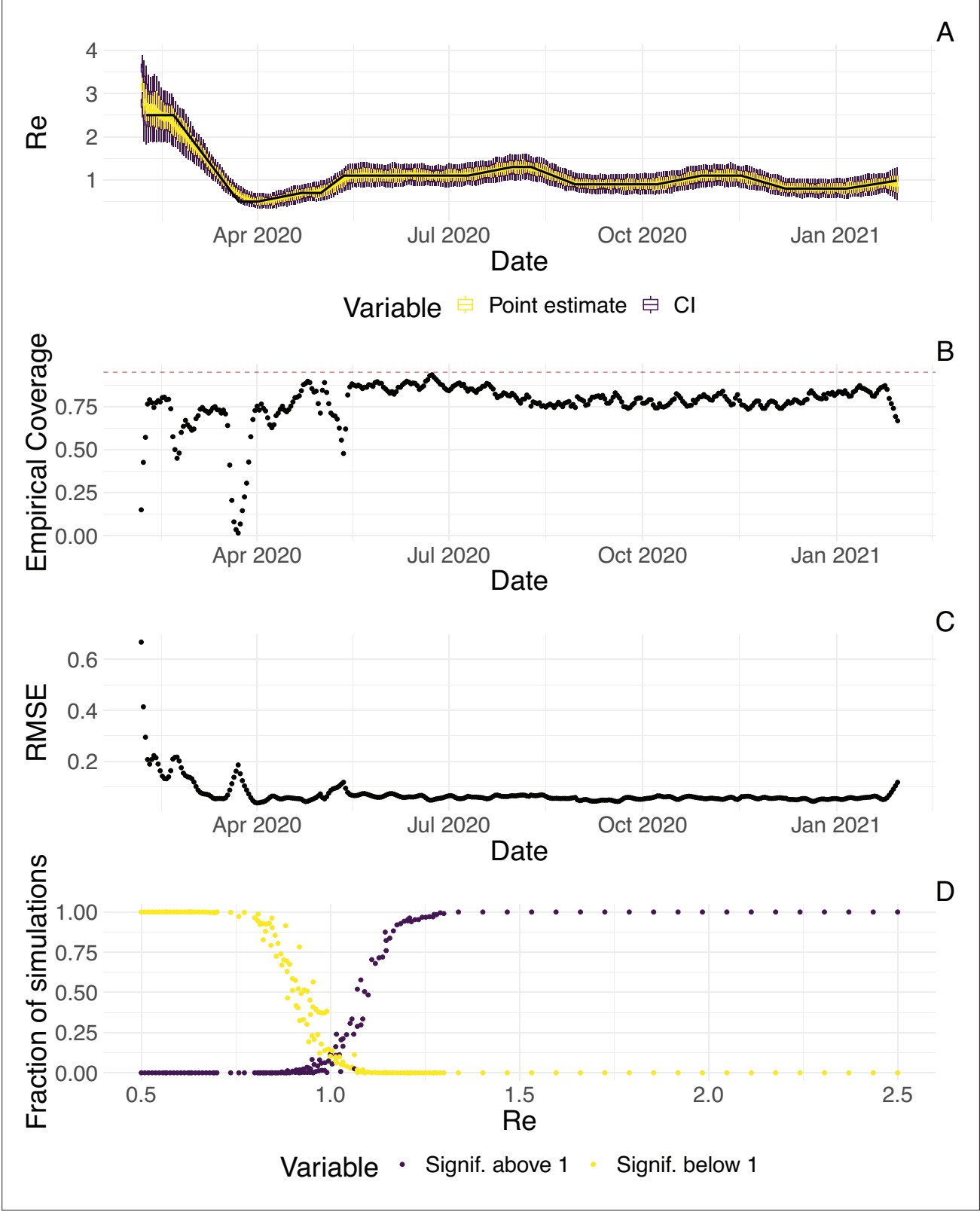

**Figure 1.** Evaluation of the pipeline on simulated data. (**A**) The specified $R_e$ trajectory (black line) was used to stochastically simulate 100 trajectories of observed cases. From each trajectory, we estimated $R_e$ (yellow boxplots) and constructed a 95% confidence interval (purple boxplots of the lower/upper endpoint). (**B**) Fraction of simulations for which the true $R_e$ value was within the 95% confidence interval. The dashed red line indicates the nominal 95% coverage. (**C**) Root mean squared relative error for every time point. (**D**) Fraction of simulations for which $R_e$ is estimated to be significantly above or below one, depending on the true value of $R_e$.

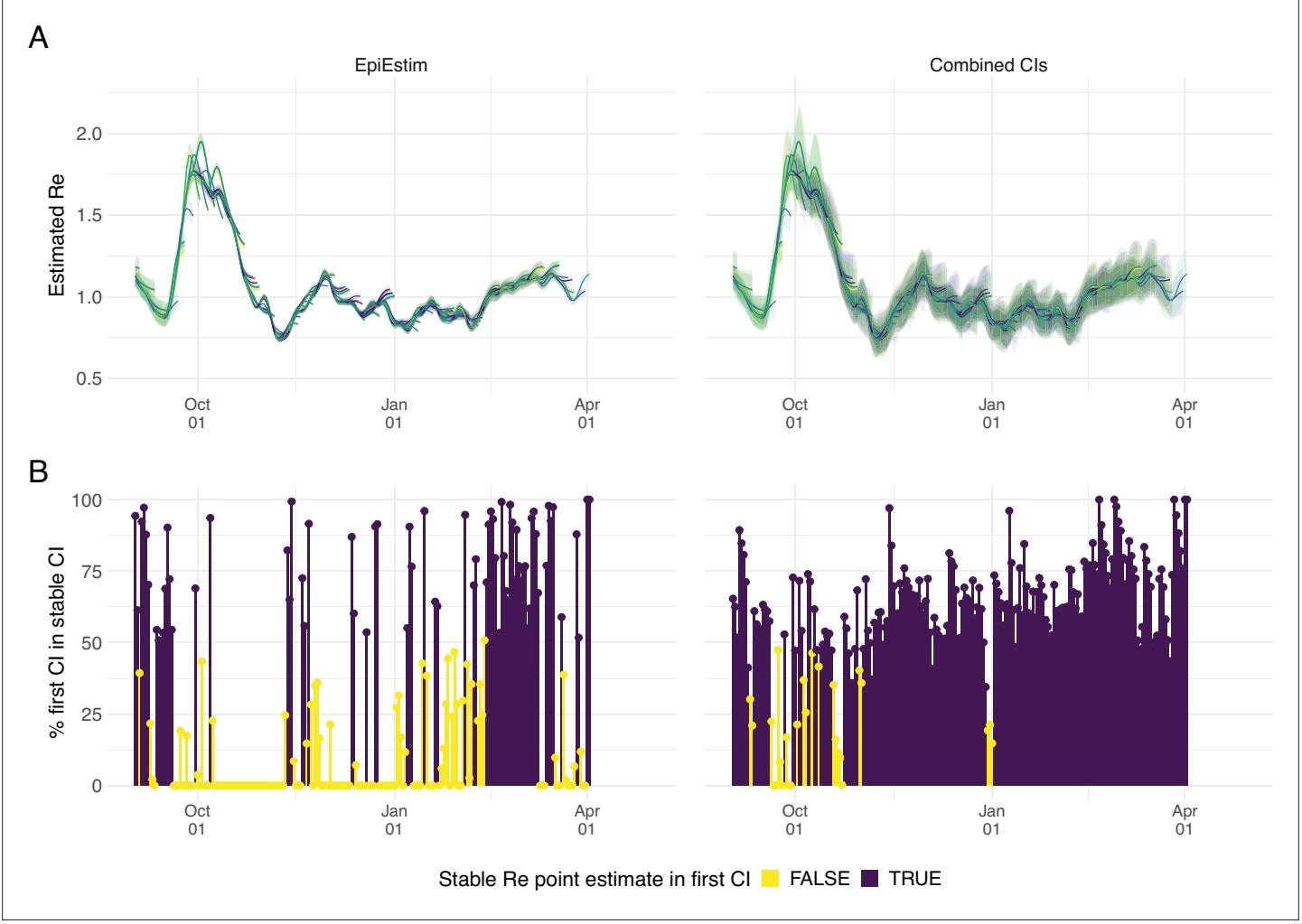

**Figure 2.** Stability of the Swiss $R_e$ estimates based on confirmed COVID-19 cases, upon adding additional days of observations. (**A**) Line segments correspond to 3 weeks of estimates made with the same input data (e.g. data up to December 1st). The segments were assigned an arbitrary colour for ease of distinction. For each day, $R_e$ estimates and associated 95% confidence intervals (CIs) are overlaid, from the first possible estimate for that day up to estimates including 3 additional weeks of data. The latter, always the left end of a line segment, corresponds to the stable estimate. (**B**) Percentage of the first estimated CI that is contained in the stable CI based on data from 30 April 2021. This percentage was calculated as the width of the intersection of both CIs, divided by the width of the first CI. The colour indicates whether the stable $R_e$ estimate was contained in the first reported CI. In both rows, the left column shows uncertainty intervals from EpiEstim on the original data, and the right our improved 95% CIs. Both columns use the same pipeline, and differ only in the construction of the uncertainty intervals.

## Stability of the $R_e$ estimates in an outbreak monitoring context

As our $R_e$ estimates for SARS-CoV-2 were directly policy relevant in Switzerland, we investigated their stability as new data becomes available (up to 21 additional days of data; *Figure 2*). With each new day of incidence data, it becomes possible to estimate $R_e$ for an additional day in the past. The most recent available $R_e$ estimate will be delayed with respect to the observed case incidence by at least the median delay from infection to observation to ensures sufficient information is available in the data (and can be delayed further if required; see below). As time passes and more observations become available, one can estimate more recent values of $R_e$. Importantly, the estimated $R_e(t)$ for a day $t$ is updated whenever additional data is added. This means that the $R_e(t)$ estimates initially change with each passing day before they settle on a long-term, stable value. In the analysis of Swiss COVID-19 case data in *Figure 2B*, $R_e$ estimates for Sept. 1 2020 to April 1 2021 based on data up to April 30 2021 are referred to as 'stable $R_e$ estimates'. During rapid changes in the real $R_e$, the initial estimates for $R_e(t)$ can occasionally under- or overshoot the long-term stable value. However, with our improved 95% confidence intervals (CI), the percentage of the first estimated CI for $R_e(t)$ that is contained

within the stable CI is substantially improved compared to purely EpiEstim-based uncertainty intervals (*Figure 2*). This difference is particularly striking during periods of high case incidence (e.g. October 2020), when the EpiEstim uncertainty interval is very narrow.

We complemented this analysis on empirical data with an assessment of the stability of $R_e$ estimates on synthetic data. Using the set-up described before, we simulated 4 $R_e$ scenarios: a constant trend, a slow increase, a slow decrease, and an abrupt decrease in $R_e$. Contrary to before, we estimated $R_e$ repeatedly, adding an additional day to the simulation in each iteration (*Appendix 4—figure 1*). For each scenario, we compare 'raw' $R_e$ estimates to trajectories for which the 4 most recent $R_e$ estimates were removed. This analysis shows that the last few $R_e$ estimates can lay outside of the stabilized confidence interval, in particular when the real $R_e$ trend is increasing. Instead, the truncated trajectories appear more stable as their most recent estimate has already been consolidated over 4 days. This highlights a trade-off between timeliness and accuracy when publishing $R_e$ estimates. On our online dashboard we present truncated $R_e$ estimates for Swiss cantons since these estimates were directly policy-relevant in Switzerland.

## Detailed data allows more precise analysis: the example of Switzerland

When detailed epidemiological data about individual cases is available (in the form of a line list), the precision of our method can be increased by relaxing the assumptions that (i) distributions of delays between infection and observation do not change through time and (ii) outbreaks occur in populations that are isolated at the country-level. In particular, we collaborated with the Federal Office of Public Health (FOPH) in Switzerland to relax these assumptions and further refine the monitoring of the Swiss SARS-CoV-2 epidemic.

The FOPH line list contains information on the delays between onset of symptoms and reporting - of a positive test, hospitalisation or death - for a substantial fraction of the reported cases. For each of these three types of case report, we estimate the time-varying empirical delay distribution for the delay from infection to reporting. We use this time-varying distribution as input to the deconvolution step, instead of the fixed delay distribution from the literature which is used for countries without an available line list (for details see Materials and Methods section 4.3). Each delay distribution is thus tailored to the specifics of the Swiss population and health system. Since each distribution varies through time, it reflects changes caused by e.g. improved contact tracing or overburdened health offices (see *Appendix 4—figure 2*; Appendix 1). Whenever available in the FOPH line list, we use the symptom onset date of patients as the date of observation and thus only deconvolve the incubation period to obtain a time series of infection dates. This was most relevant until early 2021, after which the date of symptom onset was rarely recorded anymore. For most days, the effect of these modifications on the $R_e$ point estimates is slight (*Appendix 4—figure 3*; shown for confirmed case data), yet the difference for a particular day can be as big as 20%.

Using FOPH data on the fraction of cases infected abroad, we can correct our $R_e$ estimate for imported cases. This is especially important in phases during which the local epidemic is seeded from abroad, and local transmission occurs at a low rate relative to case importation (*Appendix 4—figure 4*). This correction, which relies on EpiEstim (*Cori et al., 2013*), treats imported cases as pure infectors, and not infectees. It further assumes that imported cases transmit at the same rate as local cases. When this is not the case, e.g. when strict quarantines are imposed on travellers or travellers have more contacts compared to the rest of the population, and when a large fraction of all cases are imported, this can bias results (*Tsang et al., 2021*). Additionally, since we do not have data on the number of cases infected in Switzerland that are "exported" to other countries, we cannot correct for exports. Thus, the estimated $R_e$ value corrected for imports is a lower bound for the $R_e$ estimate which would be obtained if we could account for the location of infection of all cases detected in Switzerland or exported out of the country.

## Comparison with existing methods

During the COVID-19 pandemic, research groups and local public health authorities around the world developed and used methods to estimate the effective reproductive number (*Abbott et al., 2020a*; *Scott et al., 2020*; *Hamouda, 2020*; *Richter et al., 2020*). To put our method into context, we compare against EpiNow2, a prominent R-package to estimate (*Abbott et al., 2020a*), and the official $R_e$ estimates for Germany (developed by the *Robert Koch-Institut, 2020*; RKI *Hamouda, 2020*) and

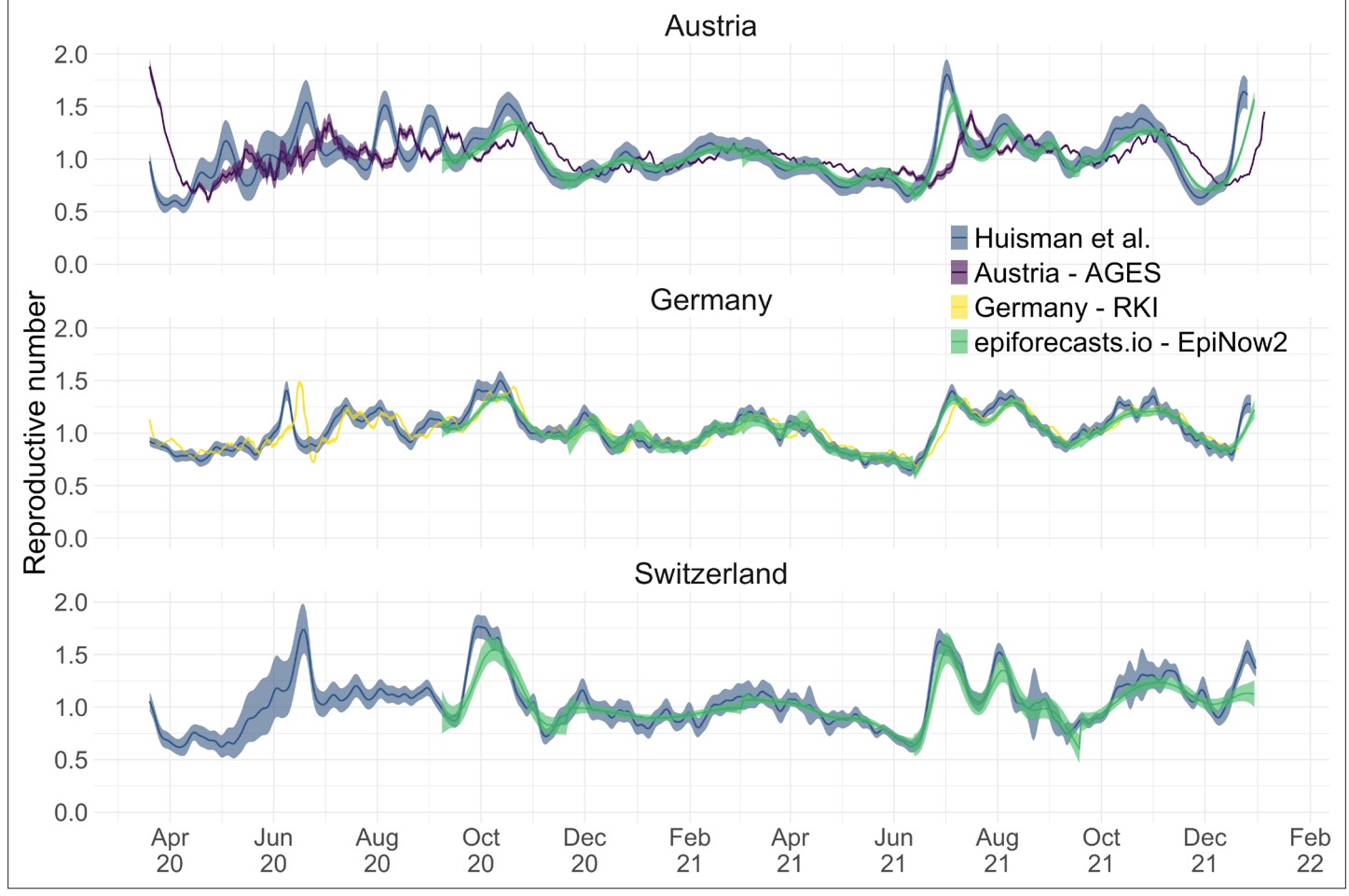

**Figure 3.** Comparison of published $R_e$ estimates for three countries. Point estimates are presented with a solid line and 95% confidence intervals are presented as coloured ribbons.

Austria (developed by the Austrian Agency for Health and Food Safety; AGES *Richter et al., 2020*). A detailed comparison of the structure and features of these methods —with the addition of the epidemia R-package (*Scott et al., 2020*)— can be found in *Supplementary file 2*. To the best of our knowledge, our method is the only one that can account for variations through time in delay distributions and combine symptom onset data with case data.

We compiled publicly available $R_e$ estimates for Switzerland, Austria and Germany from these research groups and institutions (*Figure 3*; *Abbott et al., 2022*; *Heiden, 2021*; *TU Graz AGES, 2021*). In this case the underlying truth is unknown, yet we can compare how well the point estimates and confidence intervals correspond between different methods. Both the RKI and AGES publish very narrow confidence intervals, which are unlikely to accurately capture the uncertainty around the estimates (similar to *Figure 2*). In addition, the trend of AGES estimates appear shifted closer to the present than our estimates (*Figure 3*). This is because AGES applies EpiEstim directly to observed case data, thus indirectly assuming that case confirmation occurs on the day of infection. The estimates from EpiNow2 and ours follow a qualitatively similar trend, although the EpiNow2 estimates are smoother and there is a small lag between both estimates, likely due to differences in the specified observation delay distribution. An in-depth comparison exploring the accuracy and stability of estimates produced by all available methods lies beyond the scope of this work, but would certainly be beneficial to give a full picture of the state-of-the-art of $R_e$ estimation methods.

## Monitoring $R_e$ during the COVID-19 pandemic

We developed an online dashboard (https://ibz-shiny.ethz.ch/covid-19-re-international/) on which we present daily-updated results of this $R_e$ estimation method applied to COVID-19 case data from 170

countries (*Figure 4*). For most countries, we include multiple observation sources, such as daily incidence of COVID-19 cases and deaths, and, when available, hospital admissions. We estimate $R_e$ separately from each incidence type and make the estimates available for download, as an open resource for other researchers and the general public alike.

The online app allows for comparison through time within a single country, between multiple observation traces, and between multiple countries. The data download further allows users to put these estimates in relation to external covariates such as mobility, weather, or behavioural data. The map view enables comparison across larger geographical areas and additionally reports the cases per 100'000 inhabitants per 14 days. We additionally show the Oxford Stringency Index and vaccination coverage for context (*Hale et al., 2021*; *Roser et al., 2020*).

## The effect of lockdowns in spring 2020 on the estimated $R_e$ of SARS-CoV-2

We assessed the association between non-pharmaceutical interventions (NPIs) and the estimated effective reproductive number $R_e$ during the early stage of the COVID-19 pandemic. We selected 20 European countries for which the reported data was free of major gaps or spikes, and for which we could estimate $R_e$ prior to the nationwide implementation of a lockdown in spring 2020. The dates of interventions were extracted from news reports (sources listed in *Appendix 6—table 2*), and 'lockdown' taken to refer to stay-at-home orders of differing intensity. Of the countries investigated, all except Sweden implemented a lockdown (19/20). Using case data, we inferred that $R_e$ was significantly above one prior to the lockdown measures in nearly all countries with a lockdown (15/19; *Table 1*). Denmark, which had a complex outbreak consisting of two initial waves, and Germany, which experienced a cluster of early cases, had an estimated $R_e$ significantly below one prior to this date. For countries with very short delays between the lockdown and the estimated date that $R_e < 1$ (e.g. Austria, Switzerland) we can not exclude the possibility that the 'true' $R_e$ may have been below 1 prior to the lockdown since our pipeline introduces smoothing to the estimates (see Appendix 2). The results are remarkably consistent across the different observation types (*Appendix 6—table 1*). However, the 95% confidence intervals tend to be wider for the estimates based on death incidence data because the number of deaths is much smaller than the number of cases, and the relative noise in observations tends to be higher.

To consider the association between NPIs and the estimated $R_e$ for countries outside of Europe, we used the stringency index (SI) of the Blavatnik School of Government (*Hale et al., 2020*) to describe the public health response in different countries (*Figure 4C*). This is a compound measure describing e.g. whether a state has closed borders, schools, or workplaces. For example, a country with widespread information campaigns, partially closed borders, closed schools, and a ban on public events and gatherings with more than 10 people would have an SI slightly above 50. As reference date, we used the date when a country first exceeded a stringency index of 50 ($t_{SI50}$). Then, we investigated whether the estimated $R_e$ was significantly above 1 prior to the reference date (i.e. the lower bound of the 95% confidence interval was above 1). We excluded countries without $R_e$ estimates before the reference date $t_{SI50}$. Worldwide, for 35 out of the 42 countries which fulfilled the criteria for inclusion (list in Appendix 6), $R_e$ was significantly above 1 prior to $t_{SI50}$. As a sensitivity analysis, we performed the same calculation with a different reference date $t_{max}$, defined as the date with the biggest increase in SI in the preceding 7 days. The results were very similar, with 38/45 countries significantly above one before $t_{max}$ (Appendix 6).

## Insights into continent-specific impacts of NPIs

To further investigate the association between non-pharmaceutical interventions and changes in the estimated reproductive number of SARS-CoV-2, we extended our analysis beyond the first 2020 epidemic wave. We included both the implementation and lifting of NPIs until May 3rd 2021 (increases and decreases in stringency). For each week and each country, we calculated the change in stringency index over the preceding week ($\Delta SI_t = SI(t) - SI(t-7)$) and the corresponding change in the estimated $R_e$ (during the same week $\Delta \hat{R}_{e,t} = \hat{R}_e(t) - \hat{R}_e(t-7)$, 1 week later $\Delta \hat{R}_{e,t+7} = \hat{R}_e(t+7) - \hat{R}_e(t)$, or 2 weeks later $\Delta \hat{R}_{e,t+14} = \hat{R}_e(t+14) - \hat{R}_e(t)$). If NPIs were effective, we expect increases in stringency to be associated with decreases in the estimated $R_e$ and vice versa. We do find such an association for increases in stringency e.g. in Europe. In Europe we also see that large increases in stringency

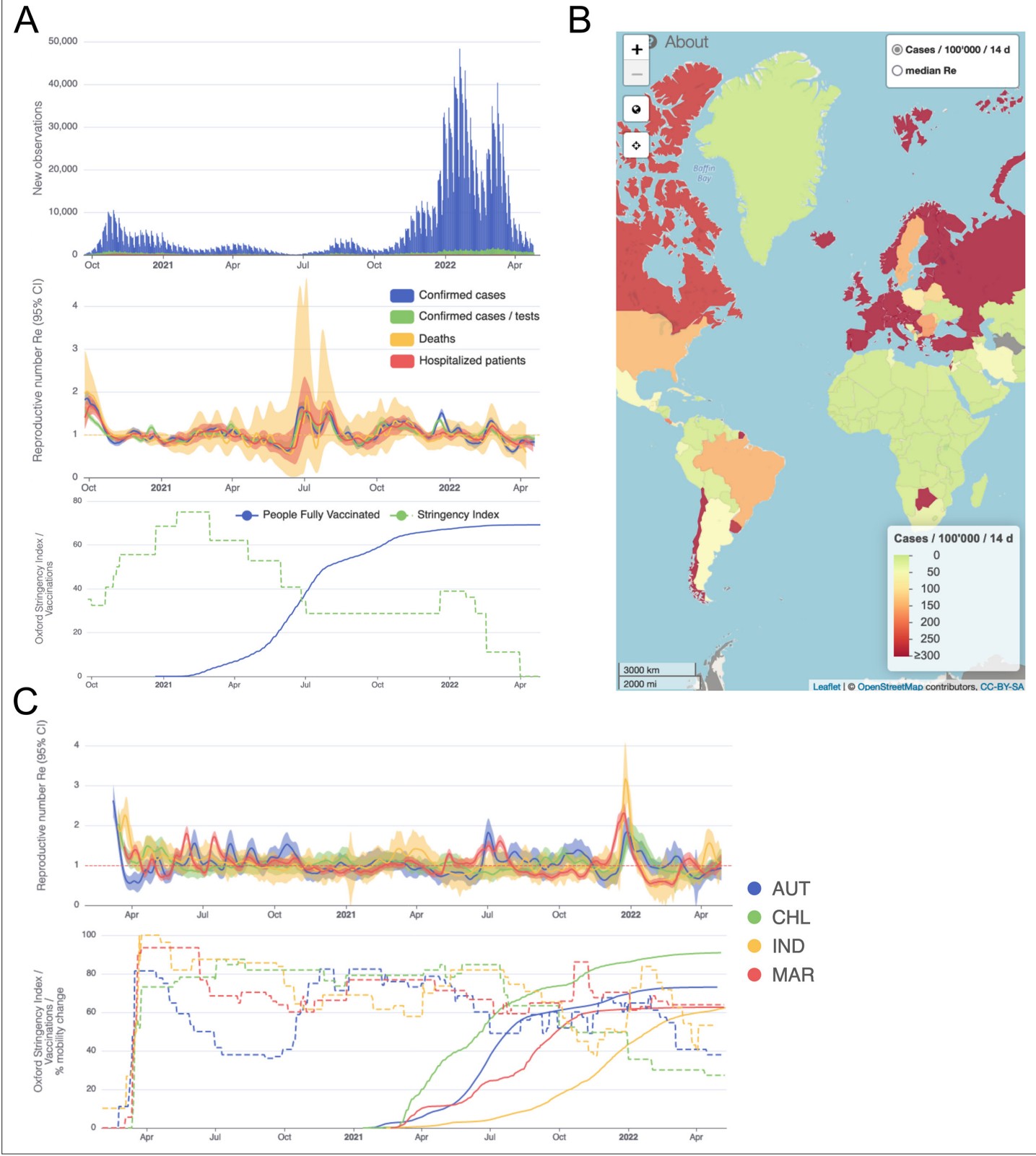

**Figure 4.** Example panels from the online dashboard. (**A**) Swiss case incidence with evidence of weekly testing patterns (top row), $R_e$ estimates with associated 95% confidence intervals from four types of observation data (middle row), and timeline of stringency index and vaccination coverage (bottom row). (**B**) World map of incidence per 100'000 inhabitants over the last 14 days. One can also display the worldwide $R_e$ estimates instead. (**C**) Comparison of $R_e$ estimates across four countries (Austria, Chile, India and Morocco), with timelines of stringency indices and vaccination coverage. All panels were extracted on May 12, 2022. Dashboard url: https://ibz-shiny.ethz.ch/covid-19-re-international.

**Table 1.** Investigating the relation between the date of 'lockdown' and the date when the estimated $R_e$ based on case reports dropped below 1.

Based on news reports, we report when a country implemented stay-at-home orders (a 'lockdown'). The column '$\hat{R}_e < 1$' indicates when the $R_e$ point estimate first dropped below 1. The column 'CI includes 1' details the corresponding time interval where the 95% confidence interval included 1. Of the investigated countries that implemented a nationwide lockdown, four (Denmark, Germany, the Netherlands, Slovenia) had 95% confidence intervals that included 1 or were below before a nationwide lockdown was implemented. The column 'Time until $\hat{R}_e < 1$' indicates the number of days between the lockdown and the date that the $R_e$ point estimate dropped below 1.

| Country | Lockdown | Re <1 | CI includes 1 | Time until Re <1 |
| --- | --- | --- | --- | --- |
| Austria | 16–30 | 20–30 | [20-03, 20-03] | 4 days |
| Belgium | 18–30 | 30–30 | [25-03, 03-04] | 12 days |
| Denmark | 18–30 | ≤10–03 | [≤10–03, 20–06] | –8 days |
| Finland | 16–30 | 01-Feb | [29-03, 30-04] | 17 days |
| France | 17–30 | 27–30 | [23-03, 07-04] | 10 days |
| Germany | 22–30 | 18–30 | [17-03, 19-03] | –4 days |
| Ireland | 27–30 | 01–100 | [04–04, 15–04] | 12 days |
| Italy | 01–300 | 18–30 | [17-03, 19-03] | 8 days |
| Netherlands | 23–30 | 01-Jan | [22-03, 10-04] | 13 days |
| Norway | 14–30 | 21–30 | [17-03, 19-03] | 7 days |
| Poland | 25–30 | 01-Jan | [31-03, 17-04] | 8 days |
| Portugal | 16–30 | 28–30 | [23-03, 15-04] | 12 days |
| Romania | 24–30 | 01-Jun | [31-03, 29-04] | 13 days |
| Russian Federation | 30–30 | 01-Apr | [01–05, 08–05] | 35 days |
| Slovenia | 20–30 | 23–30 | [≤13–03, 26–03] | 3 days |
| Spain | 14–30 | 26–30 | [25-03, 26-03] | 12 days |
| Sweden | | 01-Jan | [06–03,≥03-05-2021] | |
| Switzerland | 17–30 | 22–30 | [20-03, 22-03] | 5 days |
| Turkey | 21–30 | 01-Aug | [01–04, 13–04] | 18 days |
| United Kingdom | 24–30 | 30–30 | [28-03, 20-04] | 6 days |

are associated with larger decreases in $R_e$ 7–14 days after the change in SI. However, the association between decreases in stringency and changes in $R_e$ is heterogeneous on all continents (*Figure 5A*). This suggests that reversing non-pharmaceutical interventions had a different effect than introducing them.

We repeated the same analysis for Europe, comparing against various measures of Google mobility data (*Figure 5B*). Increased mobility in residential areas, and decreased mobility at workplaces or grocery stores is associated with decreases in $R_e$.

## Discussion

We have developed a pipeline to estimate the effective reproductive number $R_e$ of SARS-CoV-2 for both timely monitoring and retrospective investigation. We evaluated our estimates on simulated data. We showed that the inferred $R_e$ curve can be over-smoothed on simulated data, but that this disadvantage is outweighed by the increased stability of the estimates. Overall, we show that the relative error in the $R_e$ estimates is small.

During the ongoing SARS-CoV-2 pandemic, $R_e$ estimates are of interest to health authorities, politicians, decision makers, the media and the general public. Because of this broad interest and the

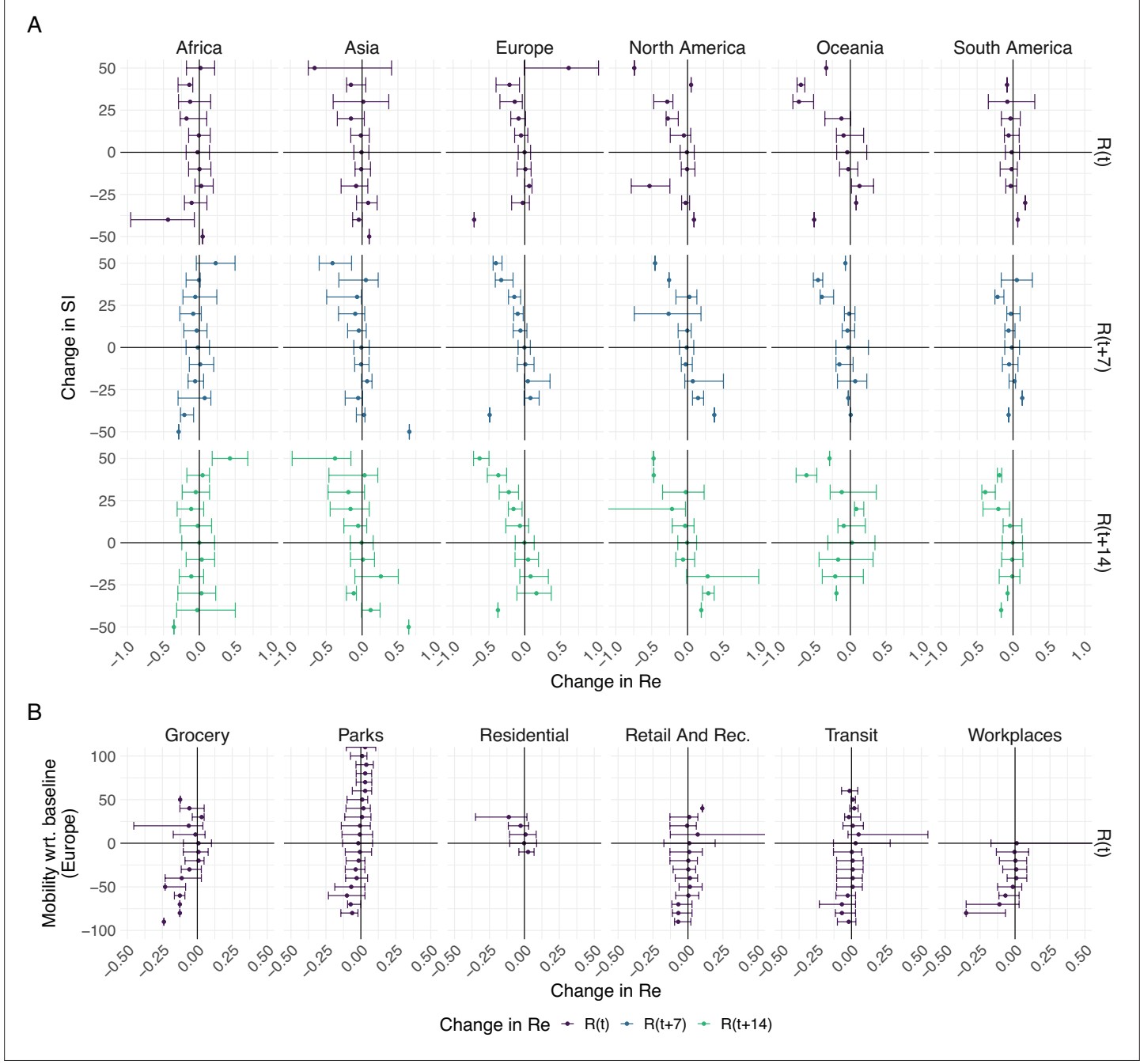

**Figure 5.** The association between the implementation or lifting of non-pharmaceutical interventions and changes in $R_e$ until May 2021. (**A**) The change in the estimated $R_e$ at the same time as ($R(t)$) or following ($R(t+7)$ and $R(t+14)$) the implementation (above x-axis) or lifting (below x-axis) of NPIs in a given week. (**B**) The change in the estimated $R_e$ related to the change in mobility in the same week. The error bars indicate the Q1 and Q3 quartiles.

importance of $R_e$ estimates, it is crucial to communicate both the results as well as the associated uncertainty and caveats in an open, transparent and accessible way. This is why we display daily updated results on an online dashboard, accessible at https://ibz-shiny.ethz.ch/covid-19-re-international/. The dashboard shows $R_e$ estimates in the form of time series for each included country or region, and a global map containing the latest $R_e$ estimates and normalised incidence. For all countries, we further display a timeline of the stringency index of the Blavatnik School of Government (*Hale et al., 2021*), and current vaccination coverage.

A unique advantage of the monitoring method we have developed is the parallel use of different types of observation data, all reflecting the same underlying infection process (*Scire et al., 2020*). Wherever we have data of sufficient quality, we estimate $R_e$ separately based on confirmed cases, hospitalisations and death reports. The advantages and disadvantages of the different observation types are discussed in the Appendix 1. Comparing estimates from several types of data is a powerful way to evaluate the sensitivity of the results to the type of observations they were derived from. More generally, the method is applicable to any other type of incidence data, such as admissions to intensive care units or excess death data. We have also extended this method to make use of daily measurements of SARS-CoV-2 viral concentrations in wastewater (*Huisman et al., 2022*). The potential limitations of our $R_e$ estimation method are discussed in detail in the Appendix 1.

Any decision to implement, remove or otherwise adjust measures aimed at infection control will be informed by epidemiological, social and economic factors (*Sebhatu et al., 2020*). We can aid this decision making process by investigating the association between adjustments of public-health measures and the estimated $R_e$. In particular, the merits of nation-wide lockdowns in the context of the COVID-19 pandemic have been heavily discussed, both in the scientific literature and the public sphere (*Flaxman et al., 2020*; *Haug et al., 2020*; *Banholzer et al., 2021*; *Soltesz et al., 2020*; *Karberg, 2020*). Analyses showing that $R_e$ estimates had dropped below 1 before the strictest measures were enforced were frequently used to claim that a lockdown was not necessary (*Karberg, 2020*). We showed that this argumentation cannot be applied universally: for 15 out of 20 European countries, we found that the estimated $R_e$ was significantly above 1 prior to the lockdown in spring of 2020. Interestingly, the result we obtain for Germany critically depends on whether we use symptom onset data, or more widely available case reports.

Extending our analysis beyond the first wave, we find differences between continents in the association between changes in the stringency of NPIs and changes in $R_e$. This could reflect differences in the speed with which lockdowns were put into practice (*Kohanovski et al., 2022*), the de facto lockdown stringency, or socio-cultural aspects (*Sebhatu et al., 2020*; *Mbow et al., 2020*). It is often argued that, especially in countries with a large informal business sector, there may be a difference between the official containment measures and those adhered to or implemented de facto (*Mbow et al., 2020*). However, for continents where we find no significant correlation, this could also be because a large fraction of NPIs were implemented at a time for which we could not estimate changes in $R_e$. Many African countries had early and strict government responses, often prior to the first detected cases. These are thought to have delayed the virus in establishing a foothold on the continent (*Mbow et al., 2020*).

Importantly, our analysis suggests that reversing non-pharmaceutical interventions may have a very different effect than introducing them. This could be because the situation is not fully reverted: due to increased public awareness, testing, contact tracing, and quarantine measures still in place. In addition, the epidemic situation - in terms of number of infected individuals - is likely different when measures are implemented or lifted.

Our analysis could be confounded by economic, social, and psychological factors motivating the implementation or release of measures. With the current stringency measures we cannot account for diversity in adherence to NPIs across geographic regions and through time. Cultural norms, defiance towards public authorities, "lockdown fatigue", and economic pressures are all among the factors that may determine whether NPIs are in fact adhered to. In addition, there is increasing evidence that weather may be a factor influencing $R_e$ through its effect on people's behaviour and on properties of the virus (*Morris et al., 2021*). In the future, our tools to estimate $R_e$ could be used to explore associations of these many factors with $R_e$ estimates, with the aim of identifying minimal sets of factors that may ensure an $R_e < 1$ for a particular location.

## Materials and methods
### Overview of the software pipeline

The software pipeline we developed allows the estimation of $R_e$ from different proxies for the infection incidence, such as the time series of confirmed cases, hospitalisations or deaths. It provides a separate estimate of the $R_e$ trajectory through time for each proxy. In a first step, we smooth the case observations and deconvolve the smoothed observations by the type-specific delay distribution to

obtain an estimate of the infection incidence time series. Second, we use the package EpiEstim to estimate the effective reproductive number Re from this infection incidence. We assess the uncertainty in the estimates using the union of a block bootstrap method, designed to account for variation in the case observations, and the credible intervals from EpiEstim.

## Smoothing the case observations

To reduce the influence of weekly patterns in case reporting data, as well as reporting irregularities, we smooth the observed incidence data prior to deconvolution. To smooth the incidence data, we use local polynomial regression (LOESS) with 1st order polynomials and tricubic weights. The smoothing parameter alpha is set such that we include 21 days of data in the local neighbourhood of each point. After smoothing, we normalise to the original total number of cases. Here we use smoothing parameter 21 because it performs best overall in our simulations. We investigated the effect of this tuning parameter in simulations, see *Appendix 5—figure 5*.

## Estimating the infection incidence through deconvolution

To recover the non-observed time series of infection incidence, we deconvolve the smoothed observed time series of COVID-19 case incidence with a delay distribution specific to the type of case detection (case confirmation, hospital admission, death). To this end, we extended the deconvolution method of *Goldstein et al., 2009*, which is itself an adaptation of the Richardson-Lucy algorithm (*Richardson, 1972*; *Lucy, 1974*) (essentially an expectation maximisation algorithm), to deal with zero-incidence case observations and time-varying delay distributions.

Formally, the method infers a deconvolved output time series $(\lambda_1, \ldots, \lambda_N)$ from an input time series $(\bar{D}_K, \ldots, \bar{D}_N)$, where $K \geq 1$ and $\bar{D}_i$ indicates the smoothed number of observations on day $i$ (e.g. confirmed cases, hospitalisations, or deaths). Let $m_l^j$ be the probability that an infection on day $j$ takes $l \geq 0$ days to be observed. If no line list data is available, $m_l^j = m_l$ and no time-variation of the delay distribution is assumed. Let $q_j$ be the probability that an infection that occurred on day $j$ is observed during the time-window of observations, i.e. is counted towards $(\bar{D}_K, \ldots, \bar{D}_N)$. Then:

$$q_j = \sum_{l=K-j}^{N-j} m_l^j. \tag{1}$$

Let $E_i$ be the expected number of observed cases on day $i$, for a given infection incidence $(\lambda_k)$:

$$E_i = \begin{cases} \sum_{j=1}^{i} \lambda_j m_{i-j}^j & \text{for} \quad K \geq i \geq N \\ 0 & \text{for} \quad 0 < i < K. \end{cases} \tag{2}$$

The deconvolution algorithm uses expectation maximisation (*Dempster et al., 1977*) to find a final infection incidence estimate, which has the highest likelihood of explaining the observed input time series. To do so, it starts from an initial guess of the infection incidence time series $\Lambda^0 = (\lambda_1^0, \ldots, \lambda_N^0)$, used to compute $E_i^0$ according to *equation 2*, and updates the estimate in each iteration $n$ according to the following formula:

$$\lambda_j^{n+1} = \frac{\lambda_j^n}{q_j} \cdot \sum_{i=K}^{N} \frac{m_{i-j}^j \bar{D}_i}{E_i^n}. \tag{3}$$

The iteration proceeds until a termination criterion is reached. Here, we follow Goldstein et al. and iterate until the $\chi^2$ statistic drops below 1 (*Goldstein et al., 2009*):

$$\chi^2 = \frac{1}{N - K + 1} \sum_{i=K}^{N} \frac{(E_i^n - \bar{D}_i)^2}{E_i^n}, \tag{4}$$

or 100 iterations have been reached.

Convergence is typically fast and the stopping criterion based on the $\chi^2$ statistic is reached in a few iterations. Due to the smoothing prior to deconvolution, this is the case for the vast majority of the empirical data we analyzed. In some cases, e.g. when the observed incidence is especially noisy,

convergence is slower and the threshold of 100 iterations is reached (on 26.5.2022, this was the case for 13 of the countries analyzed).

For the initial estimate of the incidence time series $\Lambda^0$, we shift the observation time series backwards in time by the mode of the delay distribution μ *Goldstein et al., 2009*. However, this leaves a gap of unspecified values at the start and end of the time series $\Lambda_0$. Contrary to Goldstein et al., we augment the shifted time series with the first observed value ($\bar{D}_K$) on the left, and with the last observed value ($\bar{D}_N$) on the right, to avoid initialising with a zero-value anywhere. If a day is initialised with zero incidence, it will also have zero incidence in the final estimate (compare *equation (3)*), which would be a potential source of bias.

We note that the Richardson-Lucy deconvolution algorithm accounts for 'right truncation', i.e. that not all infections are observed within the given observation time window (due to delay until symptoms/reporting), through the $q_j$ indices.

## Use of line list data

When information on the time variation of delays between symptom onset and observation is available (e.g. through a line list), this can be taken into account directly during the deconvolution step. In this case, we perform the deconvolution in two separate steps: first with the time-varying empirical onset-to-observation distributions, and then with the constant-through-time incubation period distribution. For those cases where symptom onset data is available, we only deconvolve with the incubation period distribution.

The $(m_0^j, \ldots, m_{l_{max}}^j)$ time-varying delay distributions from onset of symptoms to observation are determined as follows: for each date $j$, at least 300 of the most recent recorded delays between symptom onset and observation, with onset date before $j$, are taken into account; $l_{max}$ being the highest observed delay. To avoid biases caused by the intensity of testing and reporting varying throughout the week, recorded delays are included in full weeks going in the past, until at least 00 delays are included.

As the incidence data is right-truncated, we have to fix the distribution for the reporting delay ($m_l^j$) after a certain day $j$, so that delay distributions are not downward biased for infection dates close to the present. Let ($\bar{m}_0, \ldots, \bar{m}_{l_{max}}$) be the empirical probability density function of the delay (aggregated over the entire window of observations) and $n$ the 99th percentile of this distribution ($n$ is the smallest integer for which $\sum_{i=1}^n \bar{m}_i \geq 0.99$). For infection dates $z$ that are closer to the present than $n$ (i.e. $N - z < n$, where $N$ is the index of the last available data point), we fix $(m_0^z, \ldots, m_{l_{max}}^z)$ to be equal to $(m_0^{N-n}, \ldots, m_{l_{max}}^{N-n})$.

## Estimating the effective reproductive number Re

Once we have obtained an estimate for the time series of infection incidence, we use the method developed by *Cori et al., 2013*, implemented in the EpiEstim R package, to estimate $R_e$.

Disease transmission is modelled with a Poisson process. At time $t$, an individual infected at time $t - s$ causes new infections at a rate $R_e(t) \cdot w_s$, where $w_s$ is the value of the infectivity profile $s$ days after infection. The infectivity profile sums to 1, and can be approximated by the (discretised) serial interval distribution (*Cori et al., 2013*). The likelihood of the incidence $I_t$ at time $t$ is thus given by:

$$P(I_t | I_0, \ldots, I_{t-1}, R_e(t)) = \frac{(R_e(t)\Lambda_t)^{I_t} e^{-R_e(t)\Lambda_t}}{I_t!}, \tag{5}$$

$$\text{where} \qquad \Lambda_t = \sum_{s=1}^t I_{t-s} w_s. \tag{6}$$

The $R_e$ inference is performed in a Bayesian framework, and an analytical solution can be derived for the posterior distribution of $R_e(t)$ (see *Cori et al., 2013*; Web Appendix 1). We choose a gamma distributed prior on $R_e(t)$ with mean 1, and standard deviation 5.

For the gradually-changing $R_e$ estimates, we assume $R_e$ is constant over a sliding window of 3 days ($\tau = 3$ in EpiEstim), i.e. the reported $R_e$ estimate for day $T$ summarises the average $R_e$ over a 3 day period ending on day $T$. In addition to these smooth estimates, we provide step-wise estimates of $R_e$ on our dashboard. In the step-wise analysis, $R_e$ is assumed to be constant on a number of intervals spanning the entire epidemic time window. These intervals are determined by dates at which public health interventions were implemented, altered, or lifted. All results reported here are based on the

smooth $R_e$ estimates. In both cases, we use the mean of the posterior distribution of $R_e$ as the point estimate.

## Estimating the uncertainty intervals

To account for the uncertainty in the case observations, we construct 95% bootstrap confidence intervals for $R_e$. We first re-sample case observations as follows: given the original case observations $D_t, t = K, \ldots, N$, we apply the LOESS with smoothing parameter 21 days on the log-transformed data $log(D_t + 1)$ to obtain the smoothed value $\hat{\mu}_t$ and additive residuals $e_t$. Here we use log-transformation to stabilise the variance of the residuals because it is the best overall choice among the transformations we tried. We compare to the commonly used square root transformation in *Appendix 5—figure 1*.

After obtaining the residuals $e_t$, we resample them to get bootstrap residuals $e_t^*$ and obtain the bootstrap case observations by

$$D_t^* = \max(\exp(\hat{\mu}_t + e_t^*) - 1, 0). \tag{7}$$

We now discuss how we obtain a set of bootstrapped residuals $e_t^*$, $t = K, \ldots, N$. Since the empirical residuals of most countries are autocorrelated (see *Appendix 5—figure 2*), we use an overlapping block bootstrap. Specifically, given the original residuals $(e_K, \ldots, e_N)$, we start by taking a random block of $b$ consecutive residuals, which we denote by $(e_1^{*1}, \ldots, e_b^{*1})$. To account for weekly patterns in the case observations, we need to match the day of the week to the original case observations. That is, we keep the longest subvector $(e_{m_1}^{*1}, \ldots, e_b^{*1})$ of $(e_1^{*1}, \ldots, e_b^{*1})$ such that $e_{m_1}^{*1}$ has the same day of the week as $e_K$ (e.g., both correspond to Fridays). We then randomly take a new block of $b$ consecutive residuals, which we denote by $(e_1^{*2}, \ldots, e_b^{*2})$. We keep its longest part $(e_{m_2}^{*2}, \ldots, e_b^{*2})$ such that $e_{m_2}^{*2}$ has the day of the week that follows on that of $e_b^{*1}$ (e.g. if $e_b^{*1}$ corresponds to a Tuesday, then $e_{m_2}^{*2}$ must correspond to a Wednesday). We then glue these two sampled blocks together to get $(e_{m_1}^{*1}, \ldots, e_b^{*1}, e_{m_2}^{*2}, \ldots, e_b^{*2})$. We repeat this process of adding blocks until the length of the re-sampled residuals is at least as large as that of the original residuals. If it is longer, we simply cut off the last part of the re-sampled residuals so that its length is the same. Finally, we re-index the re-sampled residuals as $(e_K^*, \ldots, e_N^*)$. We present a concrete example in Appendix 2.

Choosing an optimal block size $b$ for the block bootstrap method is generally difficult. To capture week effects, we need a block size of at least 7. We tried different sizes and found that $b = 10$ tended to work well in various simulation settings (*Appendix 5—figure 6*).

Given a set of bootstrap case observations $(D_K^*, \ldots, D_N^*)$, we apply our method to obtain an estimate for $R_e(t)$. For ease of notation, we now denote this by $\hat{\theta}^*(t)$. By repeating the above steps 100 times, we obtain $\hat{\theta}_1^*(t), \ldots, \hat{\theta}_{100}^*(t)$. Then, we construct a Normal based bootstrap confidence interval for each time point $t$ by:

$$[\hat{\theta}(t) - q_z(1 - \tfrac{\alpha}{2})\widehat{sd}(\hat{\theta}^*(t)), \ \hat{\theta}(t) + q_z(1 - \tfrac{\alpha}{2})\widehat{sd}(\hat{\theta}^*(t))], \tag{8}$$

where $\hat{\theta}(t)$ denotes the estimated $R_e(t)$ based on the original case observations, $q_z(1 - \tfrac{\alpha}{2})$ denotes the $1 - \tfrac{\alpha}{2}$ quantile of the standard normal distribution, and $\widehat{sd}(\hat{\theta}^*)$ denotes the empirical standard deviation of $\hat{\theta}_1^*(t), \ldots, \hat{\theta}_{100}^*(t)$. In this paper, we aim at a confidence interval level of 95%, so $\alpha = 0.05$. We use the Normal based bootstrap interval because we found that it performed best overall with respect to coverage in our simulations, when compared to other common choices like quantile and reversed-quantile bootstrap confidence intervals (*Appendix 3—figure 3*).

The above bootstrap method implicitly assumes that the variance of the residuals $e_t$ is constant over time $t$ and does not depend on the value of the log-transformed data $\log(D_t + 1)$. This assumption roughly holds when the case incidence is high. During periods of low case incidence (e.g. deaths or regional data in summer 2020 in Switzerland), this assumption is no longer appropriate. Therefore, to be conservative and rather err on the side of too wide uncertainty intervals, we also consider the credible interval of $R_e$ which is obtained by taking the 0.025 and 0.975 quantiles from the posterior distribution of $R_e(t)$ using EpiEstim based on the original data $(D_K, \ldots, D_N)$. The final reported interval is then the union of the credible interval and the 95% bootstrap confidence interval. Based on our experience, the credible interval is typically wider during periods of very low case incidence and will

**Table 2.** Gamma distributions used in the pipeline: serial interval, incubation period, and the delay distributions assumed for each observation type.

| Distribution | Mean (days) | SD (days) | Reference |
|---|---|---|---|
| Serial interval | 4.8 | 2.3 | *Nishiura et al., 2020* |
| Infection to onset of symptoms | 5.3 | 3.2 | *Linton et al., 2020* |
| Onset of symptoms to case confirmation | 5.5 | 3.8 | *Bi et al., 2020* |
| Onset of symptoms to hospital admission | 5.1 | 4.2 | *Pellis et al., 2021* |
| Onset of symptoms to death | 15.0 | 6.9 | *Linton et al., 2020* |

then be reported. But at high case numbers, the bootstrap confidence interval will tend to be much wider than the credible interval and will be the reported one.

Finally, we point out that the choices of transformation, block size, smoothing parameters and type of bootstrap confidence interval in this paper might not be universal. The best choice can be different for different data sets (e.g., data sets from different countries).

## Data

We gather case incidence data directly from public health authorities. Whenever accessible, we rely on data from local authorities. Otherwise, we use data from 'Our World in Data' since the European Centre for Disease Control (ECDC) has stopped its daily updates (December 2020) (*Roser et al., 2020*; *European Centre for Disease Prevention and Control (ECDC), 2022*). A table summarising the incidence data sources is available in *Supplementary file 1*. Information on the start and end of interventions, or major changes in testing policy, are obtained from media reports and the websites of public health authorities. The stringency index of the Blavatnik School of Government is accessed from their publicly available github repository (*Hale et al., 2020*). The vaccination coverage is taken from 'Our World in Data' (*Roser et al., 2020*).

We parametrise the discretised infectivity profile $w_s$ using COVID-19 serial interval estimates from the literature (*Nishiura et al., 2020*). For a review of published serial interval estimates, see Griffin et al. (*Griffin et al., 2020*). The incubation period is parametrised by a gamma distribution with mean 5.3 days and SD 3.2 days (*Linton et al., 2020*). For countries for which we do not have access to line list data, i.e. all except Switzerland, Germany and Hong Kong at the time of writing, we assume delays from symptom onset to observation to be gamma-distributed, with parameters taken from the literature. *Table 2* summarises the distributions used in our pipeline.

For Switzerland, Germany and Hong Kong, we use line lists to build time-varying empirical distributions on delays between symptom onset and case confirmation, hospitalisation or death. During the deconvolution step we use the empirical delay distribution of the last 300 recorded cases prior to the infection date. Moreover, for the fraction of cases for which the date of onset of symptoms is known, we use the onset date directly instead of deconvolving a delay from onset to reporting, allowing for more precise estimation of the infection date. For Switzerland, line lists contain information on which cases were infected abroad. By considering imported cases and locally-transmitted cases separately in the deconvolution step, we obtain two separate time series, one for local infections and one for imported infections. *EpiEstim* can then estimate a corrected $R_e$ that excludes infections incurred abroad from the local transmission dynamics.

For comparison between methods, epiforecasts.io $R_e$ estimates were collected from https://github.com/epiforecasts/covid-rt-estimates/blob/master/national/cases/summary/rt.csv, accessing file versions from December 5 2020, March 15, June 24, October 1, November 10 2021 and January 10 2022. Robert Koch Institute estimates were collected from https://raw.githubusercontent.com/robert-koch-institut/SARS-CoV-2-Nowcasting_und_-R-Schaetzung/main/Nowcast_R_aktuell.csv, last accessed on January 10 2022. AGES estimates were collected from https://www.ages.at/fileadmin/AGES2015/Wissen-Aktuell/COVID19/R_eff.csv, last accessed on January 10 2022.

## Simulations

In the simulations, we start by specifying different $R_e$ trajectories. To assess a range of scenarios, we parametrise $R_e$ as a piecewise linear trajectory, where we fix its plateau values and the time-points at which its slope changes. From each $R_e$ trajectory, we stochastically simulate 100 time series of infections and their corresponding case observations. We then use our pipeline to estimate $R_e(t)$ from each of these 100 times series and compare it to the true underlying value of $R_e(t)$.

Assuming $I_0$ infected individuals on the first day, the infection incidence is simulated forward in time. The infection incidence on day $t$ is drawn from a Poisson distribution, corresponding to *equation (6)*, using the specified $R_e$ time series and the discretised serial interval for SARS-CoV-2 (*Nishiura et al., 2020*) as the infectivity profile (see *Cori et al., 2013*; Web Appendix 11). These simulated infections are convolved with the observation type-specific delay distribution (*Linton et al., 2020*) to obtain the raw observation time series $\tilde{D}_t$.

Since the raw observation time series $\tilde{D}_t$ are too smooth compared to the real data (*Appendix 5—figure 4*), we add noise to obtain our final simulated observation time series $D_t$. The additional noise accounts for aspects of the observation process that are not covered by the delay distribution, such as weekend and holiday effects, the random and occasional delay in the recording of confirmed cases, and irregular components such as confirmed cases that are imported from abroad.

To obtain a realistic noise model for $D_t$, we considered the empirical noise observed in real SARS-CoV-2 case data. For most countries the residuals are autocorrelated (*Appendix 5—figure 2*), which led us to fit ARIMA models to the observed residuals. We considered five simulation settings with different noise models obtained based on the confirmed case data from five countries (Switzerland, China, France, New Zealand, United States of America). Specifically, we first apply the LOESS smoother with smoothing parameter 21 days on the log-transformed confirmed case data to obtain additive residuals. We then chose the ARIMA model by fitting ARIMA models of various orders and assessing the resulting ACF and PACF plots of their residuals. This leads to five ARIMA models: ARIMA(2,0,1)(0,1,1), ARIMA(1,0,1)(0,0,0), ARIMA(0,0,6)(0,1,1), ARIMA(4,0,1)(1,0,0), and ARIMA(4,0,0)(0,0,0), based on the data from CHE, CHN, FRA, NZL, and USA, respectively. The final observation time series $D_t = \tilde{D}_t \cdot \exp(e_t)$, where $e_t$ is simulated from the fitted ARIMA model. We present the simulated observations with the noise model based on CHE data in *Appendix 5—figure 4*. We emphasize that the ARIMA model is only used in simulations to obtain simulated observations that look roughly realistic. Our main approach to obtain the estimated $R_e$ and the related confidence intervals does not require fitting an ARIMA model. In particular, the block bootstrap method is fully non-parametric.

In the case of time-varying delay distributions, we assume that the mean of the delay distribution decreases by a fixed amount (1/20) each day, to a minimum of 2 days (e.g. for the confirmed cases this results in a range from 5.5 to 2). When estimating with a time-varying delay distribution, we draw observations from the true distributions, similar to line list information recorded by public health authorities. To assess the added value of the deconvolution method, we further compare against a method where we estimate the infection time series by shifting the observations back by the mean of the delay distribution (termed 'fixed shift method').

To quantify the performance of our method on the simulated scenarios, we compute the root mean squared error (RMSE) at time point $j$:

$$RMSE(j) \quad = \sqrt{\frac{1}{M} \sum_{m=1}^{M} \left( \hat{R}_e(j,m) - R_e(j) \right)^2}, \tag{9}$$

where $M$ is the total number of simulations, $\hat{R}_e(j,m)$ the estimated $R_e$ and $R_e(j)$ the true $R_e$ at time $j$, for simulation $m$.

For each simulation we also compute the 95% confidence interval (CI) of our estimates across 100 bootstrap replicates. The empirical coverage indicates the fraction of simulations for which our CI includes the true $R_e$ value.

## Implementation and method availability

Daily updated results of our method on global COVID-19 data are available online on https://ibz-shiny.ethz.ch/covid-19-re-international/.

The source code of the software pipeline is openly accessible at https://github.com/covid-19-Re/shiny-dailyRe; *Angst, 2022*, and the code necessary to reproduce the figures and analyses presented in this paper is available at https://github.com/covid-19-Re/paper-code; *Huisman, 2022*.

## Acknowledgements

We thank the Federal Office of Public Health Switzerland for access to their line list data, and Jūlija Pečerska for help in finding governmental datasets on COVID-19. We thank members of the modelling group of the Swiss National Covid-19 science task force for helpful discussions, and Markus Petermann and Daniel Wyler for their comments and suggestions. This work was supported by the Swiss National Science Foundation (SNSF) through grant number 31CA30_196267 (to TS), 200021_172603 (to MHM), 310030B_176401 (to SB), and NRP72 grant 407240–167121 (to SB and TS).

## Additional information

### Competing interests

Richard A Neher: Reviewing editor, *eLife*. The other authors declare that no competing interests exist.

### Funding

| Funder | Grant reference number | Author |
| --- | --- | --- |
| Schweizerischer Nationalfonds zur Förderung der Wissenschaftlichen Forschung | 31CA30_196267 | Tanja Stadler |
| Schweizerischer Nationalfonds zur Förderung der Wissenschaftlichen Forschung | 200021_172603 | Marloes H Maathuis |
| Schweizerischer Nationalfonds zur Förderung der Wissenschaftlichen Forschung | 310030B_176401 | Sebastian Bonhoeffer |
| Schweizerischer Nationalfonds zur Förderung der Wissenschaftlichen Forschung | 407240-167121 | Sebastian Bonhoeffer Tanja Stadler |

The funders had no role in study design, data collection and interpretation, or the decision to submit the work for publication.

### Author contributions

Jana S Huisman, Jérémie Scire, Conceptualization, Data curation, Software, Formal analysis, Validation, Investigation, Visualization, Methodology, Writing – original draft, Project administration, Writing – review and editing; Daniel C Angst, Conceptualization, Data curation, Software, Validation, Investigation, Visualization, Project administration, Writing – review and editing; Jinzhou Li, Software, Formal analysis, Validation, Investigation, Visualization, Methodology, Writing – review and editing; Richard A Neher, Methodology, Writing – review and editing; Marloes H Maathuis, Conceptualization, Software, Formal analysis, Supervision, Funding acquisition, Validation, Investigation, Visualization, Methodology, Writing – review and editing; Sebastian Bonhoeffer, Conceptualization, Resources, Supervision, Investigation, Project administration, Writing – review and editing; Tanja Stadler, Conceptualization, Resources, Supervision, Funding acquisition, Validation, Investigation, Methodology, Project administration, Writing – review and editing

Author ORCIDs
Jana S Huisman http://orcid.org/0000-0002-1782-8109
Daniel C Angst http://orcid.org/0000-0002-6512-4595
Richard A Neher http://orcid.org/0000-0003-2525-1407
Sebastian Bonhoeffer http://orcid.org/0000-0001-8052-3925
Tanja Stadler http://orcid.org/0000-0001-6431-535X

**Decision letter and Author response**
Decision letter https://doi.org/10.7554/eLife.71345.sa1
Author response https://doi.org/10.7554/eLife.71345.sa2

## Additional files

### Supplementary files
• Supplementary file 1. A table summarising the sources of incidence data.

• Supplementary file 2. A table comparing the structure and features of different methods to estimate Re.

• Transparent reporting form

### Data availability
The source code of the pipeline is available at https://github.com/covid-19-Re/shiny-dailyRe (copy archived at swh:1:rev:012c2892381ed8e246843556ca6ee502f134a6e0); this includes a script to download the required incidence data from public sources. The resulting estimates (updated daily) are available at: https://github.com/covid-19-Re/dailyRe-Data. The code and data (such as case, hospitalization and death occurence data) necessary to reproduce the figures in the paper is at https://github.com/covid-19-Re/paper-code (copy archived at swh:1:rev:edbe2e7a9ca3c5183772e94909d-36fbe23e124cb). Note that for Switzerland we use delay distributions from a line list provided to us by the Federal Office of Public Health (FOPH), with one row per infected individual including patient information. Due to privacy concerns, we are not allowed to share this original data. To obtain access to these records, interested individuals should contact the FOPH directly.

The following dataset was generated:

| Author(s) | Year | Dataset title | Dataset URL | Database and Identifier |
| --- | --- | --- | --- | --- |
| Angst D, Chen C | 2022 | dailyRe-Data | https://github.com/covid-19-Re/dailyRe-Data | GitHub, dailyRe-Data |

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

## Appendix 1

### Observation types and the influence of testing

Here, we briefly discuss the benefits and potential biases of the three types of observations we used. The most commonly used proxy for infection incidence is the incidence of confirmed cases. It is the least indirect way of observing infection events. However, it generally assumes that (i) the proportion of infected individuals that is tested, and (ii) the distribution of the delay between infection and testing are constant through time. Unfortunately, these assumptions do not generally hold.

As long as the sampling proportion is constant throughout the considered time period, the $R_e$ estimates of EpiEstim are not affected by under-sampling (*Cori et al., 2013*). During the COVID-19 epidemic, many countries initially restricted testing to only severe cases, before switching to a more extensive testing effort after curbing the first epidemic wave and ramping up testing capacity (*Roser et al., 2020*). Changes in testing strategy as well as bottlenecks in testing capacity can result in a varying fraction of infected individuals that are confirmed positive, and variation in the delay between infection and test confirmation. These variations can bias $R_e$ estimates, as the increase or decrease in case numbers between consecutive time points will be attributed to a change in infection incidence, rather than a change in testing.

However, it is important to note that the 'memory' inherent in the $R_e$ estimate is dictated by the infectivity profile $w_s$. An event at time $t$ which changes the proportion of true infection incidence observed per day, e.g. a change in testing policy, will bias the $R_e$ estimate for a number of days given by $w_s$ (compare Materials and Methods, *equation 6*). For SARS-CoV-2 the time needed to reach the 95% quantile of $w_s$ is 9days. We do not observe the infection incidence directly, but if the deconvolution is assumed to be perfect, the intuition for the number of days of biased $R_e$ estimates still holds.

It is further possible to investigate the influence of testing intensity, by applying the $R_e$ estimation method separately to a case incidence time series which is adjusted for the intensity of the testing effort. We have added this analysis to our online dashboard (where we show the number of confirmed cases / number of tests, normalised by the mean number of tests). However, one should note that such a normalisation does not take into account that the probability of test positivity might also change with the number of tests (e.g. by prioritising likely cases at low numbers of tests).

In contrast, the incidence of hospital admittance and deaths are likely based primarily on the severity of the symptoms, and mostly unaffected by changes in testing strategies, or the magnitude of the epidemic. This makes them valuable complementary observations of infection events (*Goldstein et al., 2009*). However, also here biases can occur. First, only a small fraction of all infections results in hospitalisation or death (a meta-analysis found an average infection fatality ratio for SARS-CoV-2 of 0.68% *Meyerowitz-Katz and Merone, 2020*). This fraction varies with the risk group of the infected population (*Meyerowitz-Katz and Merone, 2020*; *Hauser et al., 2020*; *Esteve et al., 2020*; *Yang et al., 2020*), introducing potential biases in $R_e$ estimations when outbreaks occur in particularly age-stratified settings. Also, new variants may result in different hospitalisation or fatality rates. Second, if a country's health infrastructure becomes overburdened and hospitals are forced to triage or delay admission, we expect the fraction of hospital admissions to decrease, and deaths to increase. Third, the likelihood to die from an infection may change through time as new treatment strategies are developed or if hospitals are overburdened. Additionally, guidelines used to record COVID-19 as the cause of death have changed through time for some countries (*Minder, 2020*). Lastly, the delay between infection and hospitalisation or death is expected to be longer than the delay until case confirmation, with the result that these $R_e$ estimates are less timely. One should note that these observation type specific biases could also be seen as a source of information. The types simply describe a different epidemic if very structured populations with highly different mortality rates are captured (e.g. in elderly homes).

It is important to note that all analyses here are focused on the period before vaccination mediated immunity became widespread. Since vaccinations change the fraction of infections that eventually become hospitalised or die, they may introduce temporary biases for the $R_e$ estimated from hospitalisation and death incidence. We have added the metric of vaccination coverage to the online dashboard, so one can estimate when these effects start to become important.

## Method Limitations

The $R_e$ estimation method we present in the main text relies on several assumptions. Here we highlight the limitations that occur when these assumptions are violated.

First, the geographical scale of the $R_e$ estimates is determined by the incidence data itself. The $R_e$ value calculated for a country represents an average, summarised across multiple local epidemics unfolding in different regions. $R_e$ values need not be identical in different local epidemics across a country or administrative region. In particular, in times of very low pathogen transmission, single super-spreading events can significantly increase the estimated $R_e$ of the entire country (*Lloyd-Smith et al., 2005*).

Second, in our deconvolution step we account for an incubation period and a delay from symptom onset to case observation. Implicitly, we thus assume that all reported cases come from symptomatic individuals. This is certainly true for hospitalised and deceased patients, but does not have to hold for all confirmed cases. Similar to the testing intensity (discussed in Section B.1), this would not bias our estimates as long as the fraction of asymptomatic or presymptomatic individuals is constant through time. However, the fraction of asymptomatic individuals could vary with the population structure and age-stratification. The fraction of tested presymptomatic individuals could vary with the testing strategy and the intensity of the testing effort.

Third, in our current analysis we assume a single serial interval distribution for all geographic locations and all times. However, behaviour, population contact structure, and cultural differences in dealing with infection symptoms, will cause geographic and temporal variations in the serial interval. In particular, the implementation of non-pharmaceutical interventions can significantly shorten the serial interval (*Ali et al., 2020*). Misspecification of the serial interval will lead to larger errors in $R_e$ estimates further away from one (*Gostic et al., 2020*).

Lastly, our estimates of the effective reproductive number $R_e$ are subject to changes in data reporting. There are frequent changes in the way in which public health offices update their observed incidence data: the number of variables shared (e.g. Brasil, the UK excluded testing information), their frequency (e.g. Swiss cantons moved to weekly data updates when daily numbers became low), the amount of data consolidation (i.e. to which extent values reported for a given day change in subsequent days), and what constitutes a COVID-19 case (*Minder, 2020*; *Tokyay, 2020*). These variables have all changed during the epidemic, frequently in response to political pressure or the magnitude of the local epidemic and the resulting workload at the public health offices (*Minder, 2020*). This affects the timeliness of our estimates, and can cause the estimated $R_e$ to change a bit between days.

## Appendix 2

### Discretisation of delay distributions

When approximating delay distributions by gamma distributions, we discretise these in the following fashion:

$$
\mathrm{m}_l = \begin{cases} \int_0^{0.5} f(x)\,\mathrm{d}x & l = 0 \\ \int_{l-0.5}^{l+0.5} f(x)\,\mathrm{d}x & l \in \{1, 2, \dots\}, \end{cases}
\tag{10}
$$

where $f$ is either the probability density function (p.d.f) of the gamma-distributed delay distribution, or the p.d.f of the convolution of two independent gamma-distributed delay distributions. The former applies when line list data is available, and the observed data is deconvolved with the gamma-distributed incubation period separately from the empirical delay distribution of symptom onset to observation. The latter applies whenever the observed case data is jointly deconvolved with the incubation period and the delay between symptom onset and observation.

Because the probability density function of a convolution of two independent gamma distributions does not admit a simple form in the general case, we approximate the p.d.f by drawing a million independent pairs of samples, one from each gamma distribution, summing the pairs, and computing the empirical cumulative distribution function of the sampled distribution.

### The effect of smoothing on our ability to infer when Re =1

Our LOESS smoothing roughly spreads sudden changes in $R_e$ over 20 days in the estimated $R_e$. Whether this is a substantial problem depends on the smoothness of the true $R_e$ that we are trying to estimate. Direct observations of behavioural changes, specifically changes in mobility, suggest the true $R_e$ is quite smooth: for instance it took 2–3 weeks for mobility to drop to its lowest level in response to government interventions in Switzerland (**Apple, 2021**; **Google, 2021**).

However, to get a rough feeling for the impact smoothing would have on our estimates and downstream analysis in case the true $R_e$ does change abruptly, we can use a simple analysis using a linear approximation. In the case of a step-wise change from $R_0$ to $R_1$ (with $R_0 > R_1$) at time $t_0$, the estimated smooth $R_e$ will start decreasing about 10 days prior to $t_0$, and take another 10 days after to reach the terminal value (**Appendix 2—figure 1**). When inferring the day that a certain threshold value was reached (e.g. $R_e = 1$) we will be off by a number of days $s$, dictated by $R_0$ and $R_1$. Specifically, the delay $s$ is greater if the turning point $R_{tp} = \frac{R_1 + R_0}{2}$ is further above 1, or the slope $a_{tp} = \frac{R_1 - R_0}{20}$ is closer to 0:

$$
s = (R_{tp} - 1) \frac{20}{R_0 - R_1}.
\tag{11}
$$

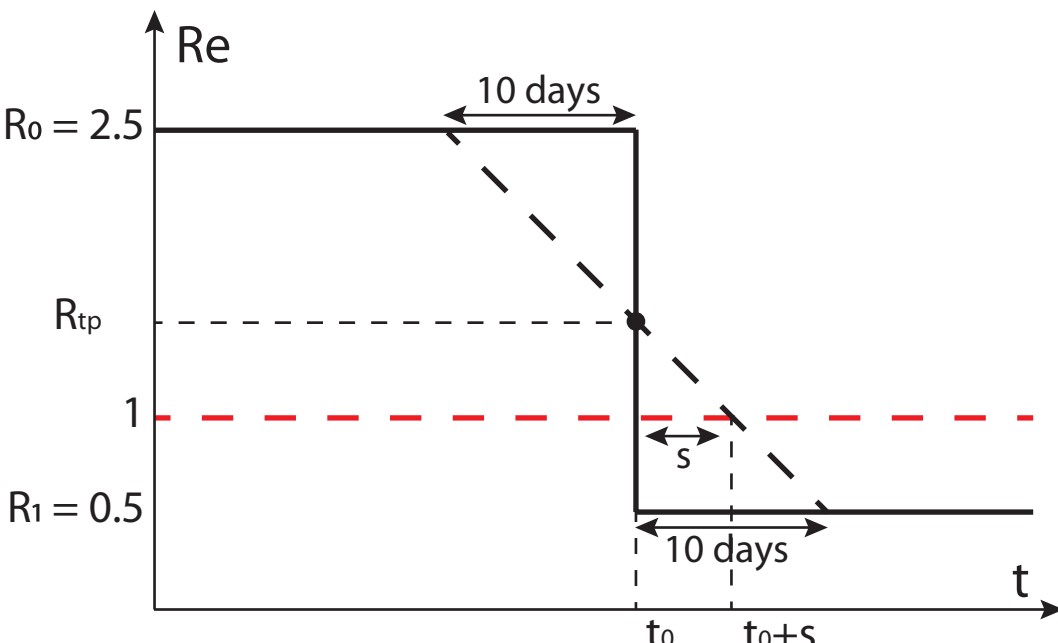

**Appendix 2—figure 1.** Schematic of the effect of smoothing on the ability to estimate when $R_e = 1$. The true $R_e$ is indicated by the black solid line, the black dashed line shows a linear approximation of the smoothed $R_e$. Instead of crossing 1 at $t_0$, this line crosses 1 at $t_0 + s$.

In *Table 1* we have listed some possible delay values, using $R_0$ values spanning the range of values reported for SARS-CoV-2 (*Alimohamadi et al., 2020*). The delay is positive if $R_0 > R_1$ and $R_{tp} > 1$, which was the case for most countries around the 1st lockdown. In general, these numbers can be considered a 'worst-case' scenario: when the true underlying $R_e$ changes more gradually than considered here, the smoothing introduced by our pipeline will have a smaller effect.

Note that these calculations specifically refer to the point estimate. The estimates may stop being significantly above the threshold already earlier, especially when the confidence interval is wide and the slope is close to 0.

**Appendix 2—table 1.** The effect of smoothing on the ability to estimate when $R_e = 1$. These values were calculated using *Equation 11*.

| $R_0$ | $R_1$ | $R_{tp}$ | $a_{tp}$(per day) | Delays (days) |
|---|---|---|---|---|
| 6.0 | 0.0 | 3.0 | −0.30 | 6.7 |
| 3.0 | 0.0 | 1.5 | −0.15 | 3.3 |
| 3.5 | 0.5 | 2.0 | −0.15 | 6.7 |
| 2.5 | 0.5 | 1.5 | −0.10 | 5.0 |
| 3.3 | 0.9 | 2.1 | −0.12 | 9.2 |
| 1.8 | 0.8 | 1.3 | −0.05 | 6.0 |

## A concrete example to illustrate the overlapping block bootstrap method

**Appendix 2—table 2.** Residuals and their corresponding days of the week.

| Day of the week | Mon | Tue | Wed | Thu | Fri | Sat | Sun |
|---|---|---|---|---|---|---|---|
| Residuals | | $e_1$ | $e_2$ | $e_3$ | $e_4$ | $e_5$ | $e_6$ |
| | $e_7$ | $e_8$ | $e_9$ | $e_{10}$ | $e_{11}$ | $e_{12}$ | $e_{13}$ |
| | $e_{14}$ | $e_{15}$ | $e_{16}$ | $e_{17}$ | $e_{18}$ | $e_{19}$ | $e_{20}$ |

*Appendix 2—table 2 Continued on next page*

*Appendix 2—table 2 Continued*

| Day of the week | Mon | Tue | Wed | Thu | Fri | Sat | Sun |
|---|---|---|---|---|---|---|---|
| | $e_{21}$ | $e_{22}$ | $e_{23}$ | $e_{24}$ | $e_{25}$ | $e_{26}$ | $e_{27}$ |
| | $e_{28}$ | $e_{29}$ | $e_{30}$ | | | | |

We illustrate the overlapping block bootstrap method with block length $b = 10$ in a small example. Let $(e_1, e_2, \ldots, e_{30})$ denote residuals corresponding to the days of the week as shown in *Table 2*.

We first randomly sample a block of 10 consecutive residuals from $(e_1, e_2, \ldots, e_{30})$. Say the sampled residuals are $(e_3, e_4, \ldots, e_{12})$. Since the first residual $e_1$ of the original residuals corresponds to a Tuesday, we then take the longest part of $(e_3, e_4, \ldots, e_{12})$ such that the first residual corresponds to Tuesday. In this case, we keep $(e_8, e_9, \ldots, e_{12})$.

The length of the sequence of re-sampled residuals is now only 5, which is less than the desired total length 30, so we have to sample again. We randomly sample a second block of 10 consecutive residuals from $(e_1, e_2, \ldots, e_{30})$. Say the sampled residuals are now $(e_{19}, e_{20}, \ldots, e_{28})$. Since the last residual $e_{12}$ of the first set of re-sampled residuals corresponds to a Saturday, we take the longest part of $(e_{19}, e_{20}, \ldots, e_{28})$ such that the first residual corresponds to a Sunday. In this case, we keep $(e_{20}, e_{21}, \ldots, e_{28})$. We then glue it to the previously sampled residuals and obtain $(e_8, e_9, \ldots, e_{12}, e_{20}, e_{21}, \ldots, e_{28})$ with length 14. Since this is less than the desired length 30, we need to sample again.

So we randomly sample a third block of 10 consecutive residuals from $(e_1, e_2, \ldots, e_{30})$. Say the re-sampled residuals are $(e_1, e_2, \ldots, e_{10})$. Since the last residual $e_{28}$ of the kept residuals corresponds to a Monday, we take the longest part of $(e_1, e_2, \ldots, e_{10})$ such that the first residual corresponds to a Tuesday. In this case, we keep $(e_1, e_2, \ldots, e_{10})$. We then glue it to the previously kept residuals and obtain $(e_8, e_9, \ldots, e_{12}, e_{20}, e_{21}, \ldots, e_{28}, e_1, e_2, \ldots, e_{10})$. Its length is 24, which is less than the original length of 30, so we keep going.

We randomly sample a fourth block of 10 consecutive residuals from $(e_1, e_2, \ldots, e_{30})$. Say the sample residuals are $(e_{17}, e_{18}, \ldots, e_{26})$. Since the last residual $e_{10}$ of the kept residuals corresponds to a Thursday, we take the longest part of $(e_{17}, e_{18}, \ldots, e_{26})$ such that the first residual corresponds to Friday. In this case, we keep $(e_{18}, e_{19}, \ldots, e_{26})$. We then glue it to the previously kept residuals and obtain $(e_8, e_9, \ldots, e_{12}, e_{20}, e_{21}, \ldots, e_{28}, e_1, e_2, \ldots, e_{10}, e_{18}, e_{19}, \ldots, e_{26})$. Note that its length is 33 which is larger than the original length of 30, so we cut the last three residuals $(e_{24}, e_{25}, e_{26})$.

This means that we obtain $(e_8, e_9, \ldots, e_{12}, e_{20}, e_{21}, \ldots, e_{28}, e_1, e_2, \ldots, e_{10}, e_{18}, e_{19}, \ldots, e_{23})$ as our set of block bootstrapped residuals.

## Appendix 3

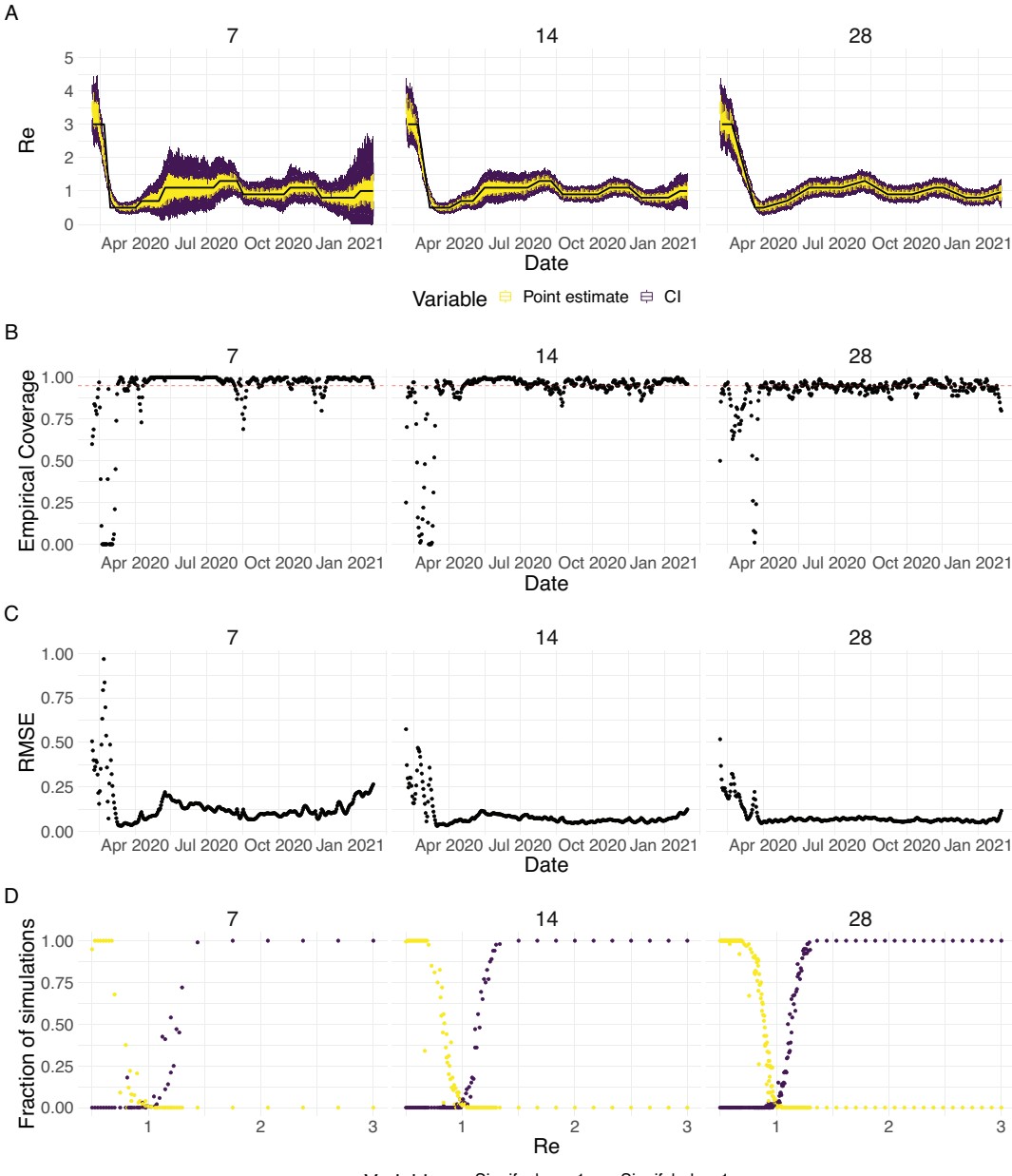

**Appendix 3—figure 1.** Performance of our method on simulated scenarios with differing slopes. (**A**) The specified $R_e$ trajectory (black line; see Methods) was used to simulate a trajectory of reported cases (with Swiss case observation noise) 100 times. From each trajectory we estimated $R_e$ (yellow boxplots), and constructed a 95% confidence interval (purple boxplots of the lower/upper endpoint). We varied the time it took to change from one $R_e$ value to the next, $t \in \{7, 14, 28\}$ (columns). Larger values of $t$ correspond to less abrupt changes. (**B**) The fraction of simulations where the true $R_e$ value was within the 95% confidence interval. The dashed red line indicates the nominal 95% coverage. (**C**) The root mean squared relative error for every time point. (**D**) The fraction of simulations where we estimate $R_e$ is significantly above or below one, depending on the true value of $R_e$. We see that the method closely tracks the true $R_e$ in all scenarios, although the error is greater for steeper slopes. In the case of steeper changes in $R_e$ the overall size of the epidemic is also smaller, which explains the larger confidence intervals.

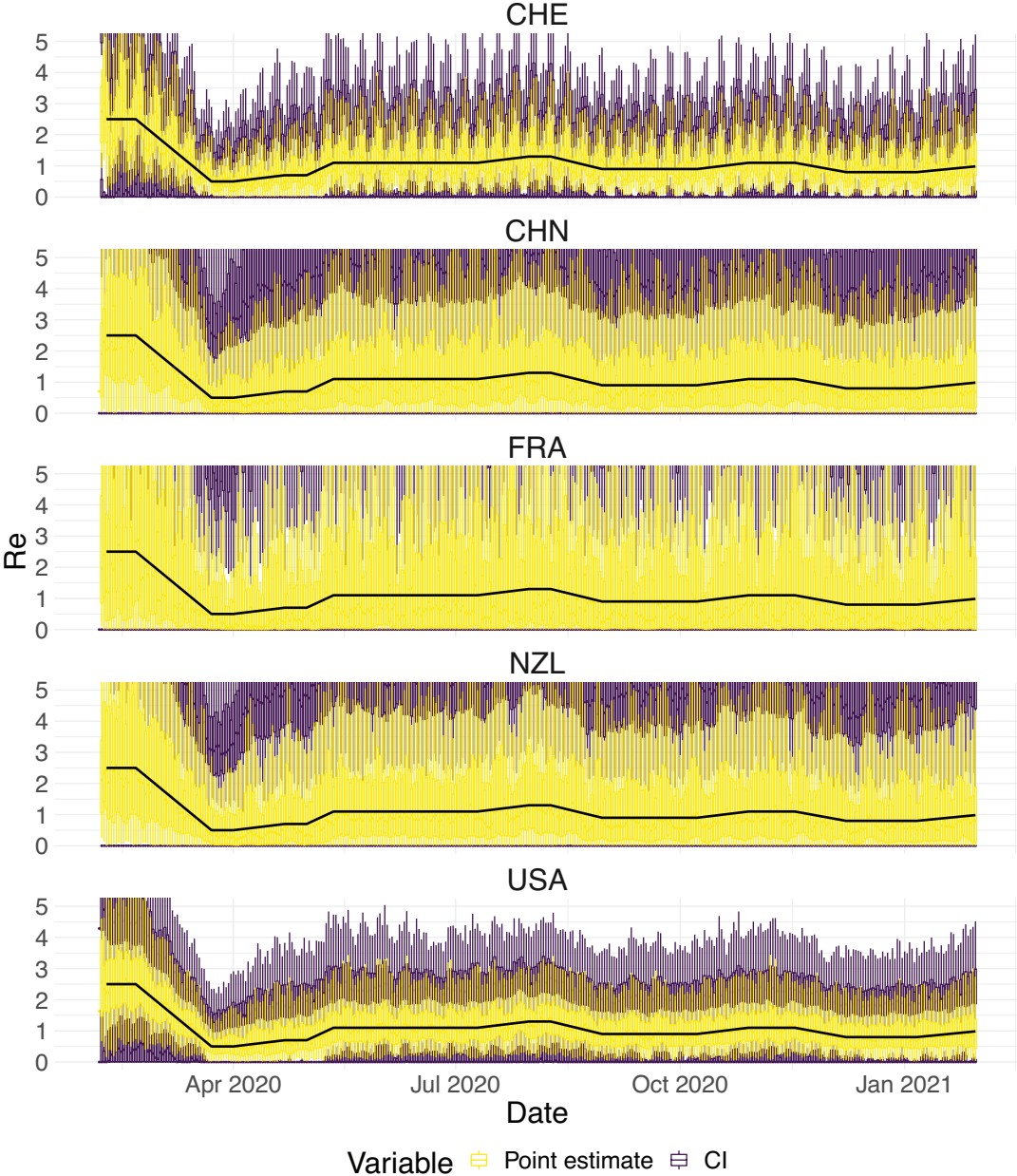

**Appendix 3—figure 2.** Performance of our method, modified to skip the smoothing step in the pipeline, on simulated scenarios with observation noise. The specified $R_e$ trajectory (black line; see Methods) was used to simulate a trajectory of reported cases (with varying country-specific noise profiles; rows) 100 times. From each trajectory we estimated $R_e$ (yellow boxplots), and constructed a 95% confidence interval (purple boxplots of the lower/upper endpoint). Contrary to our normal pipeline, the observations were not smoothed prior to the deconvolution and $R_e$ estimation. We see that the estimates are highly variable.

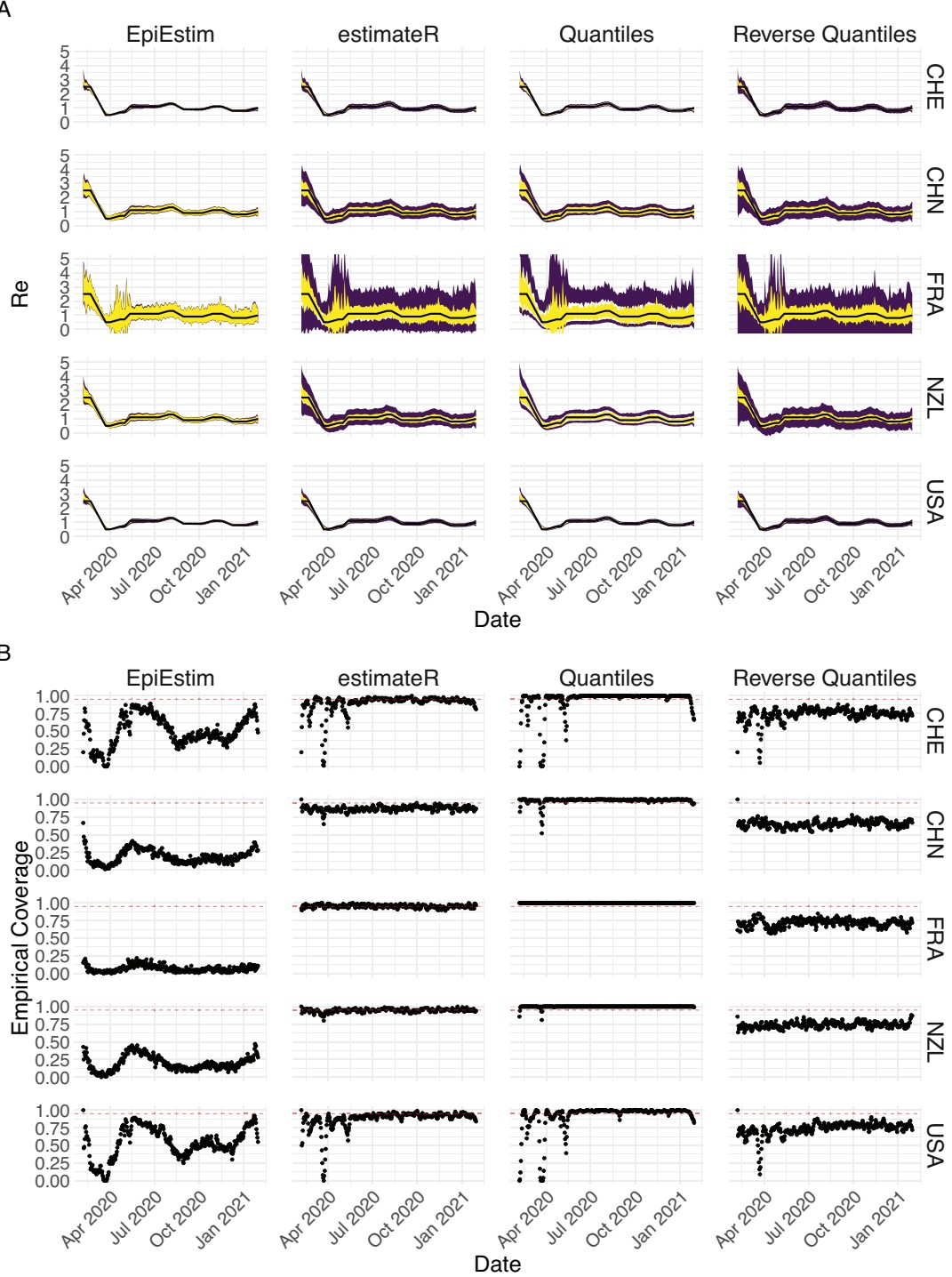

**Appendix 3—figure 3.** Performance of our method on simulated scenarios with observation noise. The columns differ in the method used to construct confidence intervals: EpiEstim reports the 95% HPD of EpiEstim on the original data, estimateR refers to our method, Quantiles and Reverse Quantiles use the 5 and 95% quantiles of the estimated $R_e$ to construct the CIs. (**A**) The specified $R_e$ trajectory (black line; see Methods) was used to simulate a trajectory of reported cases (with varying country-specific noise profiles; rows) 100 times. From each trajectory we estimated $R_e$ (yellow ribbons represent the estimated mean ± sd across 100 simulations), and constructed a 95% confidence interval (purple ribbons represent the mean ± sd of the estimated lower/upper endpoint). (**B**) The fraction of simulations where the true $R_e$ value was within the 95% confidence interval. The dashed red line indicates the nominal 95% coverage.

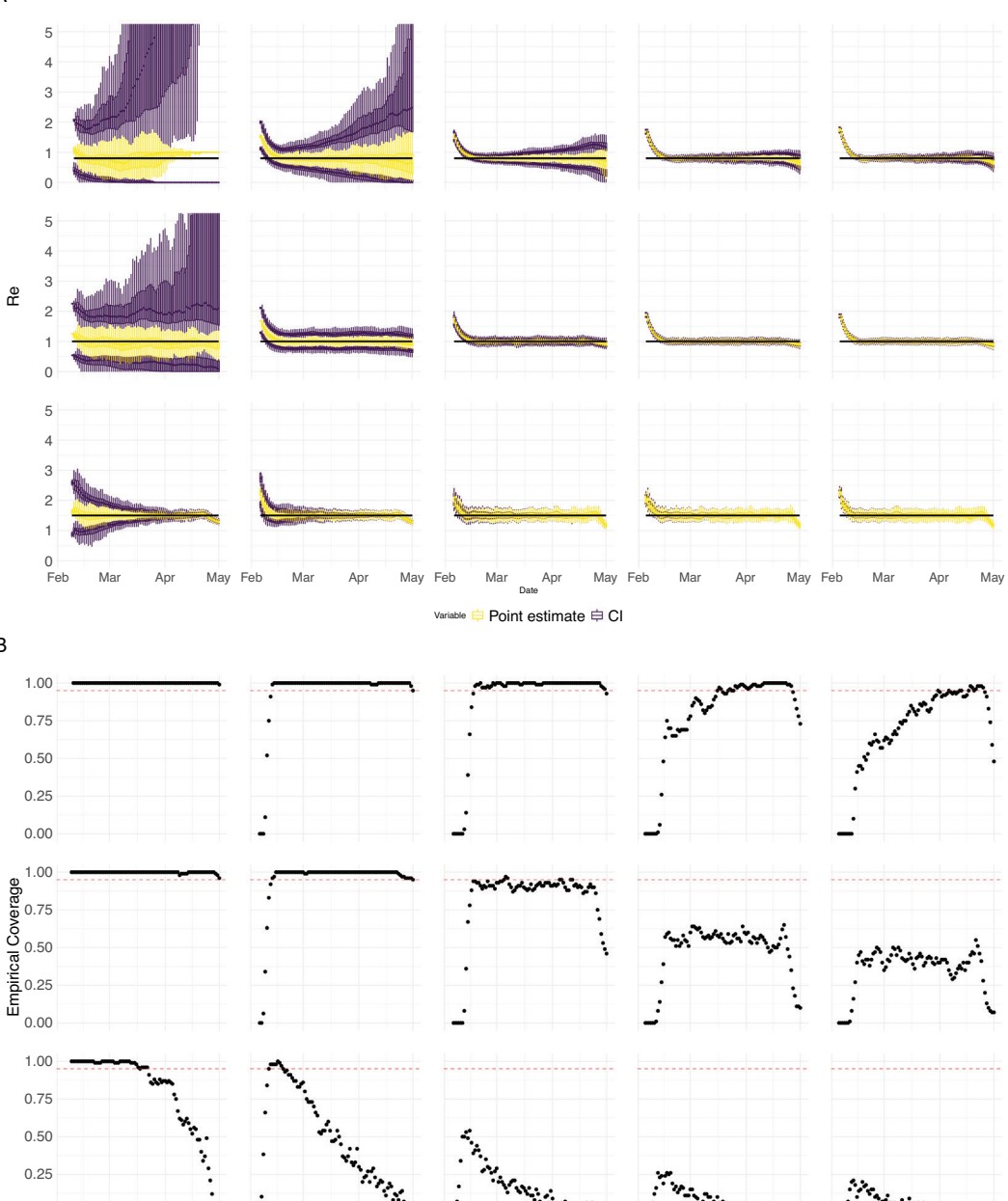

**Appendix 3—figure 4.** Performance of our method on simulated scenarios with varying population size, using confidence intervals from EpiEstim. (**A**) We specified a constant $R_e \in \{0.8, 1, 1.5\}$ value (black line; rows) to simulate a trajectory of reported cases (with Swiss case observation noise) 100 times. From each trajectory we estimated $R_e$ (yellow boxplots), and constructed a 95% confidence interval (purple boxplots of the lower/upper endpoint). The simulated scenarios had differing initial incidence of $I_0 \in \{10, 100, 1000, 5000, 10000\}$ infections per day (columns). In the top row, $R_e < 1$ so the epidemic is decreasing. In the middle row, $R_e = 1$, the epidemic is constant, and in the bottom row, $R_e > 1$, the epidemic is increasing. The bias at the start is due to the initialisation of the simulation. (**B**) The fraction of simulations where the true $R_e$ value was within the 95% confidence interval. The dashed red line indicates the nominal 95% coverage. We see that the EpiEstim coverage strongly declines with increased epidemic size.

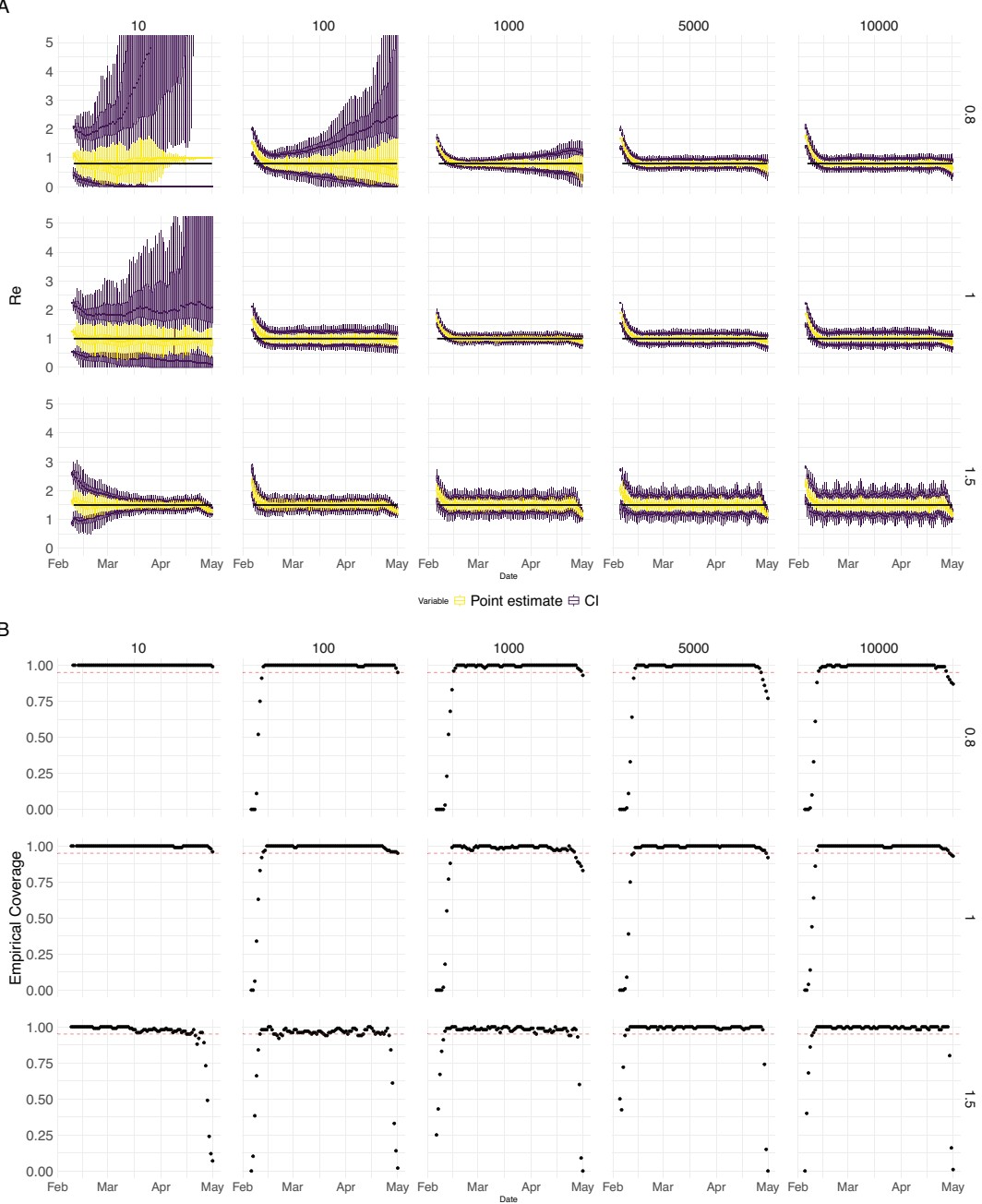

**Appendix 3—figure 5.** Performance of our method on simulated scenarios with varying population size, using the Union of EpiEstim and Block bootstrap 95% confidence intervals. (**A**) We specified a constant $R_e \in \{0.8, 1, 1.5\}$ value (black line; rows) to simulate a trajectory of reported cases (with Swiss case observation noise) 100 times. From each trajectory we estimated $R_e$ (yellow boxplots), and constructed a 95% confidence interval (purple boxplots of the lower/upper endpoint). The simulated scenarios had differing initial incidence of $I_0 \in \{10, 100, 1000, 5000, 10000\}$ infections per day (columns). In the top row, $R_e < 1$ so the epidemic is decreasing. In the middle row, $R_e = 1$, the epidemic is constant, and in the bottom row, $R_e > 1$, the epidemic is increasing. The bias at the start is due to the initialisation of the simulation. (**B**) The fraction of simulations where the true $R_e$ value was within the 95% confidence interval. The dashed red line indicates the nominal 95% coverage. We see that for a wide range of infection incidences, our 95% confidence interval is informative and covers the true value of $R_e$.

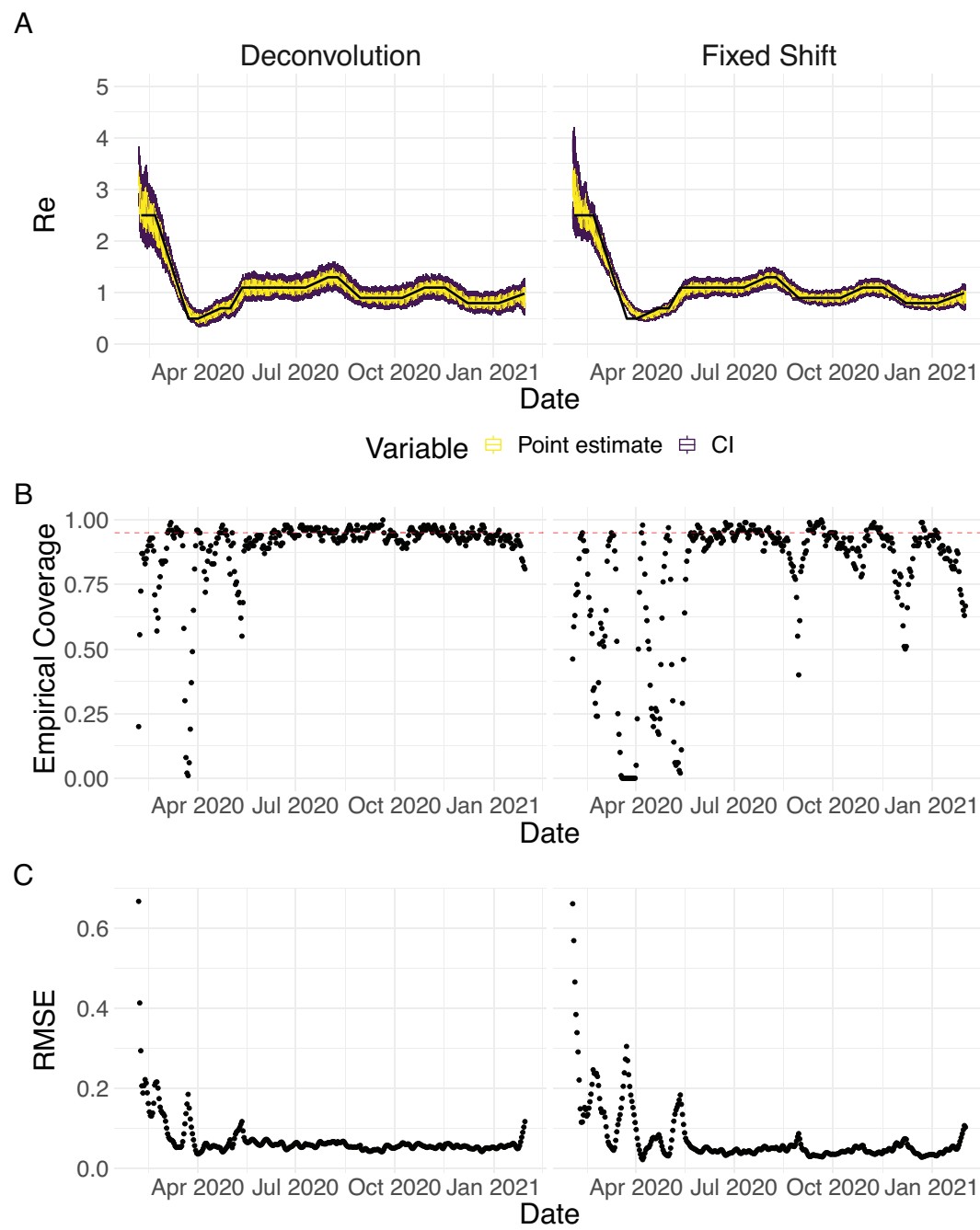

**Appendix 3—figure 6.** Performance of our method on simulated scenarios using a fixed shift versus the deconvolution to infer infection incidence. The fixed shift method shifts the observations back by the mean of the delay distribution (here assumed to correspond to confirmed cases). (**A**) The specified $R_e$ trajectory (black line; see Methods) was used to simulate a trajectory of reported cases (with Swiss case observation noise) 100 times. From each trajectory we estimated $R_e$ (yellow boxplots), and constructed a 95% confidence interval (purple boxplots of the lower/upper endpoint). (**B**) The fraction of simulations where the true $R_e$ value was within the 95% confidence interval. The dashed red line indicates the nominal 95% coverage. The average coverage in this scenario was 0.90 with deconvolution and 0.78 with the fixed shift. (**C**) The root mean squared relative error for every time point. The average (cumulative) RMSE in this scenario was 0.0706 (25.4) with deconvolution and 0.0726 (26.6) with the fixed shift.

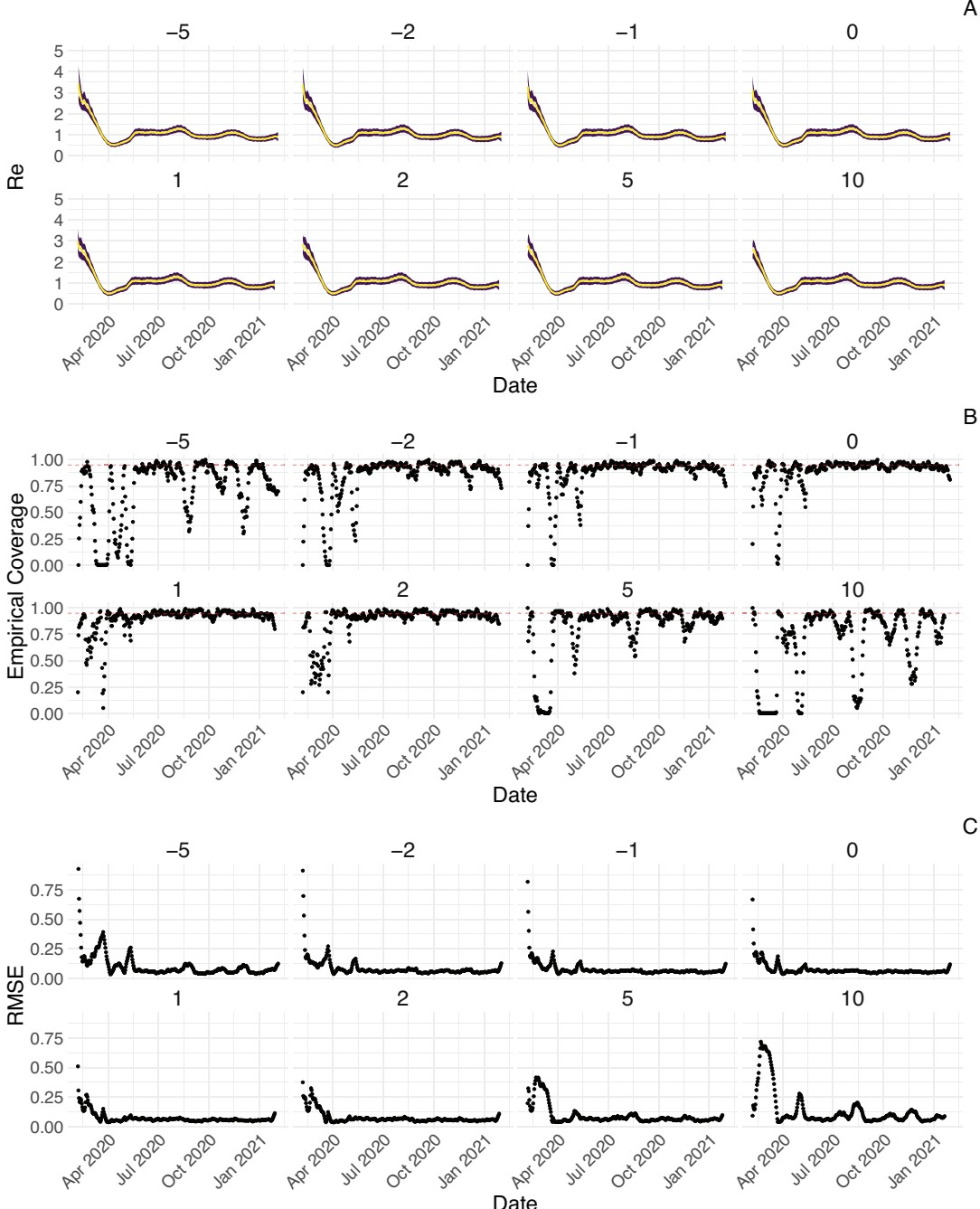

**Appendix 3—figure 7.** Performance of our method on simulated scenarios with misspecified delay distributions. When estimating $R_e$, we misspecified the mean of the delay distribution (5.5 for symptom-onset to case confirmation) by the numbers above the columns. (**A**) The specified $R_e$ trajectory (black line; see Methods) was used to simulate a trajectory of reported cases (with Swiss case observation noise) 100 times. From each trajectory we estimated $R_e$ (yellow boxplots), and a 95% confidence interval (purple boxplots of the lower/upper endpoint). (**B**) The fraction of simulations where the true $R_e$ value was within the 95% confidence interval. The dashed red line indicates the nominal 95% coverage. The average coverage in this scenario was 0.72, 0.85, 0.88, 0.90, 0.90, 0.89, 0.82, 0.69 from −5 to 10. (**C**) The root mean squared relative error for every time point. The average (cumulative) RMSE in this scenario was 0.0989 (36.0), 0.0807 (29.2), 0.0746 (26.9), 0.0706 (25.4), 0.0701 (25.2), 0.0726 (26.0), 0.0909 (32.3), 0.134 (47.0) from −5 to 10.

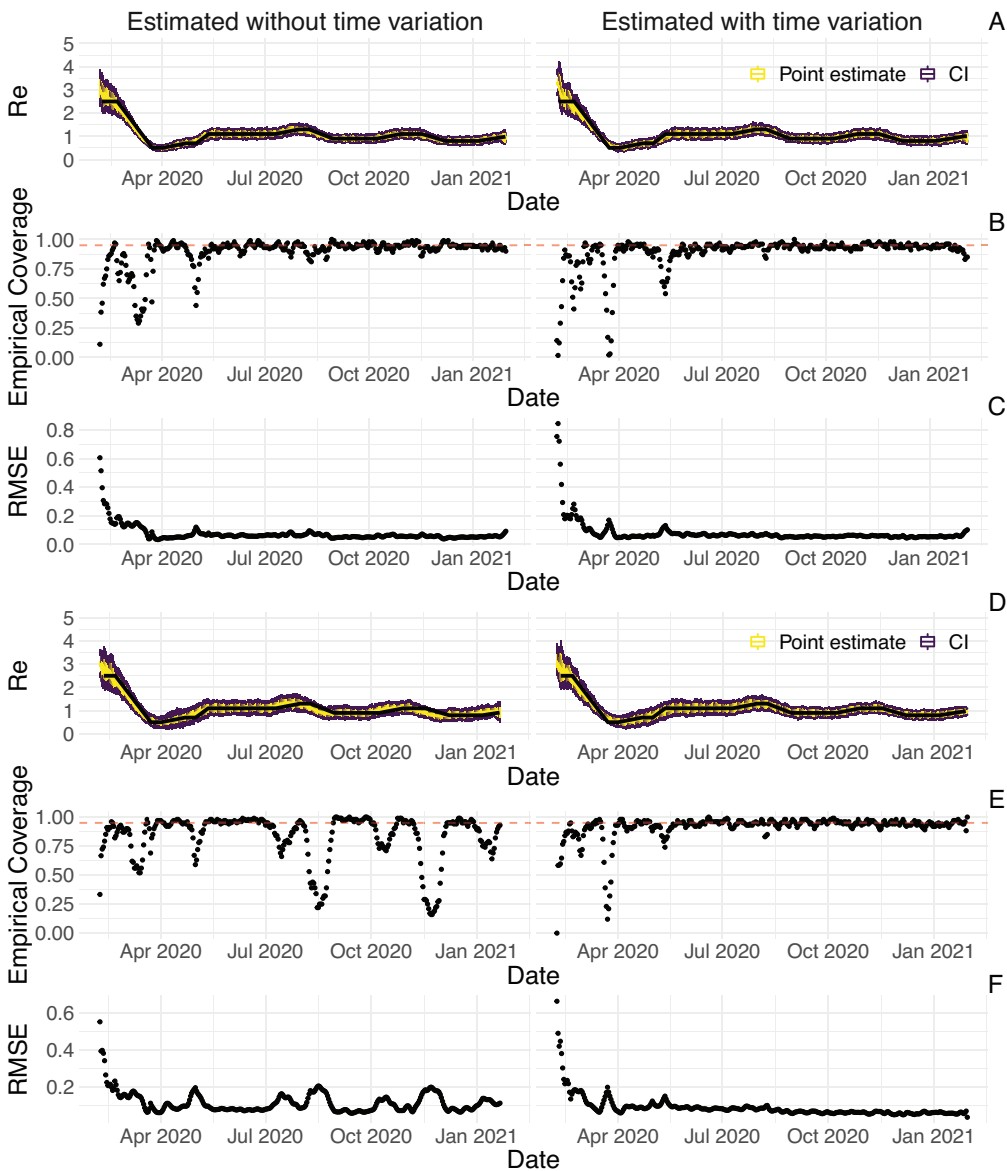

**Appendix 3—figure 8.** Performance of our method on simulated scenarios with time-varying delay distributions. The observations were simulated with a time-varying delay distribution for (**A,B,C**) confirmed cases, or (**D,E,F**) deaths (see Methods), and then estimated with (right column) or without (left column) taking the time-varying distributions into account. (**A, D**) The specified $R_e$ trajectory (black line; see Methods) was used to simulate a trajectory of reported cases or deaths (with Swiss case observation noise) 100 times. From each trajectory we estimated $R_e$ (yellow boxplots), and a 95% confidence interval (purple boxplots of the lower/upper endpoint). (**B, E**) The fraction of simulations where the true $R_e$ value was within the 95% confidence interval. The dashed red line indicates the nominal 95% coverage. For the cumulative cases, the average coverage in this scenario was 0.89 without and 0.89 with time variation. For the deaths, the average coverage was 0.83 without and 0.92 with time variation. (**C, F**) The root mean squared relative error for every time point. For the cumulative cases, the average (cumulative) RMSE was 0.0728 (26.2) without and 0.0783 (28.5) with time variation. For the deaths, the average (cumulative) RMSE was 0.113 (39.5) without and 0.0861 (30.8) with time variation.

## Appendix 4

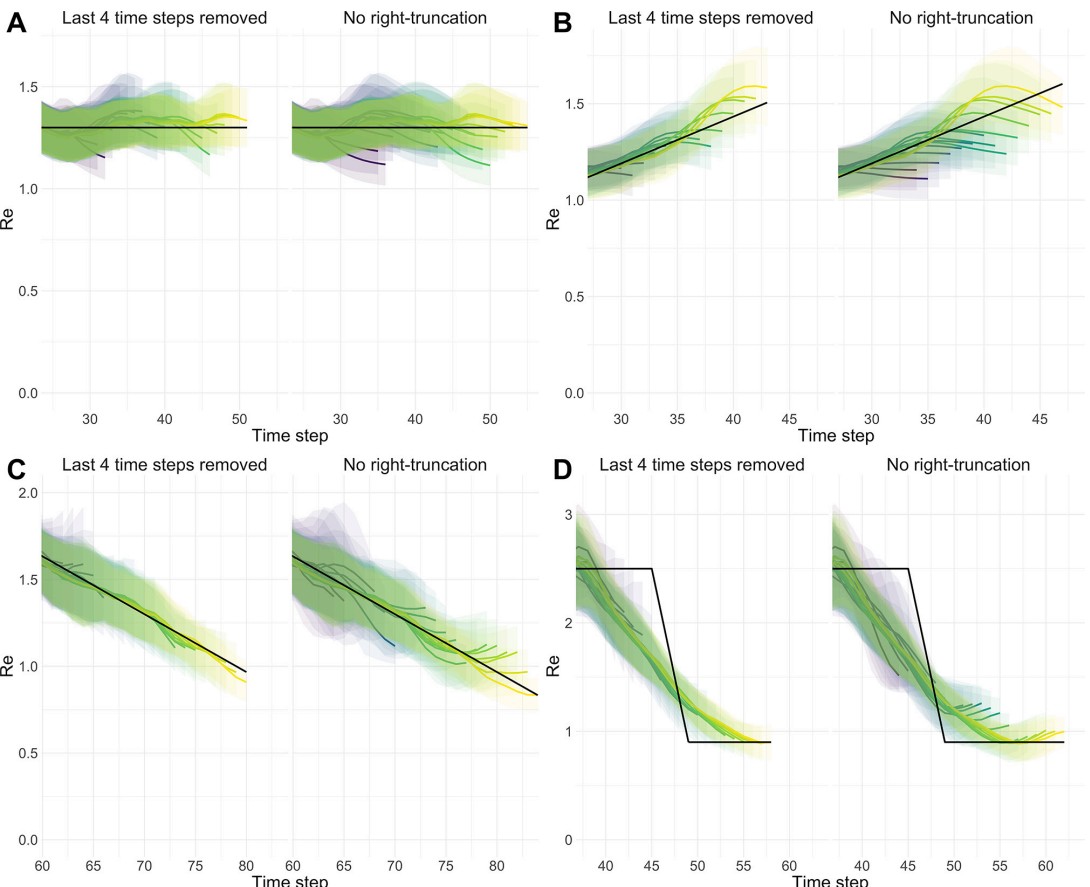

**Appendix 4—figure 1.** Stability of $R_e$ estimates at present. We estimated $R_e$ through time repeatedly on 4 scenarios. With each new iteration, we added one new data point at present. Each $R_e$ trajectory is presented with its own colour. Purple trajectories are iterations for which the last known case data point was furthest in the past, yellow trajectories are trajectories for which the last known case data point was closest to the present. In each panel and for each $R_e$ trajectory, 100 simulation replicates were aggregated. The median of mean estimates are presented with lines and medians of upper and lower bounds of 95% confidence intervals are shown with translucent ribbons. For each scenario, two panels are presented. Each time the right panel correspond to raw estimates and the left panel corresponds to the same estimates with the last 4 $R_e$ values removed from each $R_e$ trajectory. (**A**) Stable $R_e$. (**B**) Gradual increase in $R_e$. (**C**) Gradual decrease in $R_e$. (**D**) Abrupt decrease in $R_e$.

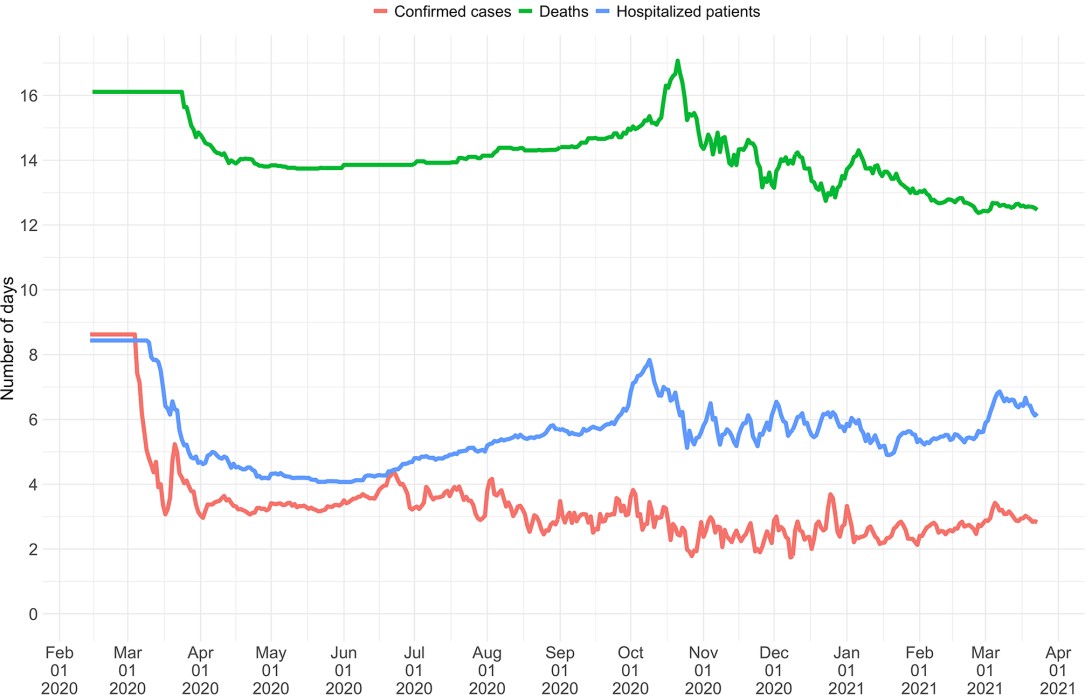

**Appendix 4—figure 2.** Mean delay in Switzerland between onset of symptoms and reporting. For each date, the mean is taken over the last 300 reports with known symptom onset date, based on line list data from the FOPH. For early dates, before 300 reports are available, the mean is taken over the first 300 reports.

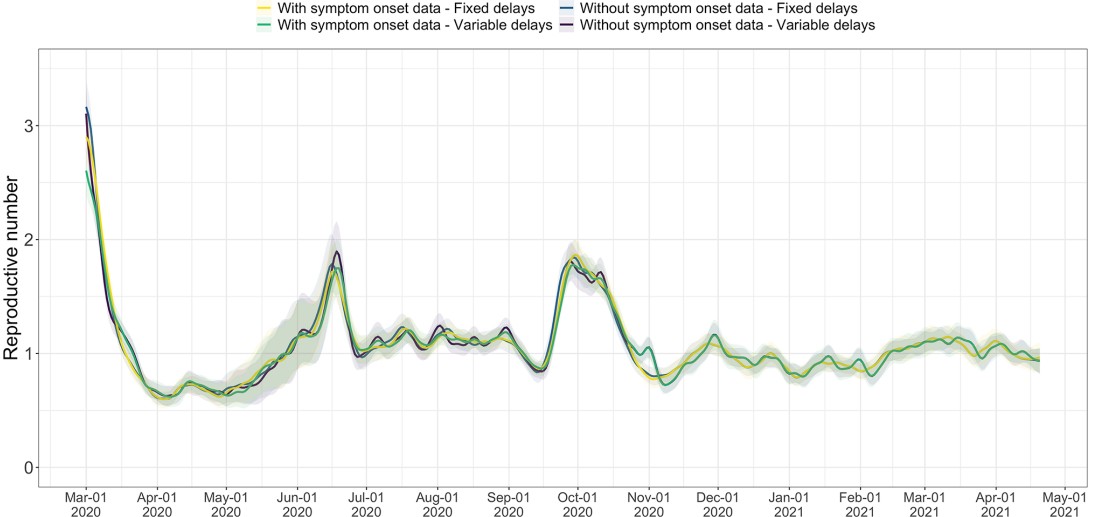

**Appendix 4—figure 3.** Comparison of the $R_e$ estimates with or without accounting for known symptom onset dates and for time-variability on reporting delays. The comparison is based on time series of confirmed cases in Switzerland, from line list data provided by the FOPH. Both the inclusion of known symptom onset dates and of the time-variability of reporting delay distributions have an effect on the Re estimates, in particular for early estimates in this case. The fraction of cases with known symptom onset date has drastically reduced since November 2020, hence the overlap in curves with and without symptom onset data for later dates. For each trajectory the point estimate is shown with a line, and the translucent ribbon indicates the 95% confidence interval.

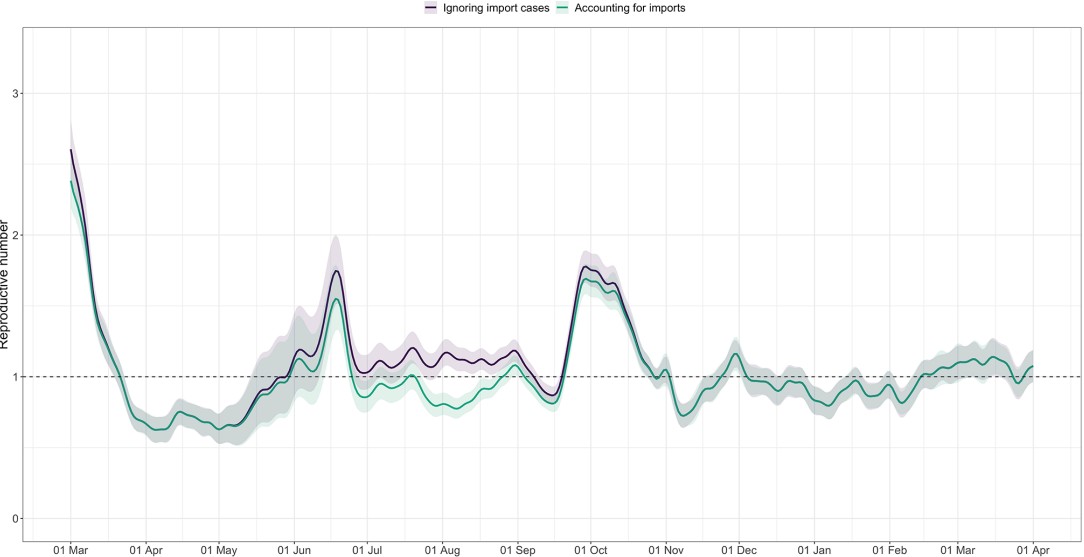

**Appendix 4—figure 4.** Effective reproductive number estimates with or without accounting for known imports. The comparison is based on time series of confirmed cases in Switzerland, from line list data provided by the FOPH. The analysis ignoring imports is unbiased if the number of imports equals the number of exports. Since the analysis accounting for imports is not accounting for exports, the results are a lower limit for the effective reproductive number. Very few imported cases were reported since November 2020, hence the complete overlap in the curves after that date. For each trajectory the point estimate is shown with a line, and the translucent ribbon indicates the 95% confidence interval.

## Appendix 5

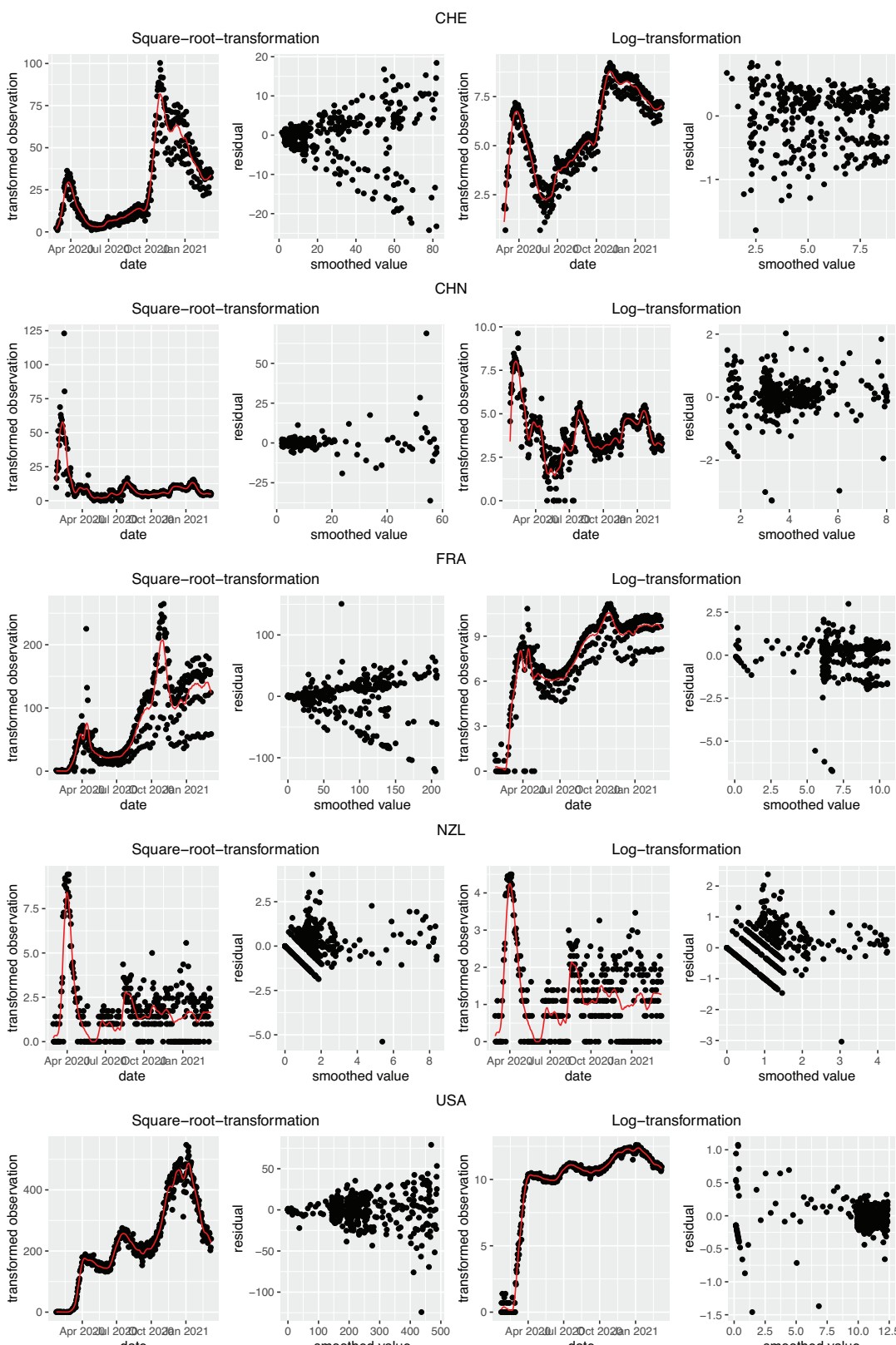

*Appendix 5—figure 1 continued on next page*

*Appendix 5—figure 1 continued*
**Appendix 5—figure 1.** Square root and log transformations to stabilise the variance of residuals. Each row corresponds to the results of observations from Switzerland (CHE), China (CHN), France (FRA), New Zealand (NZL) and the United States (USA), respectively. The first and last two plots correspond to the result of square root transformation and log transformation, respectively.

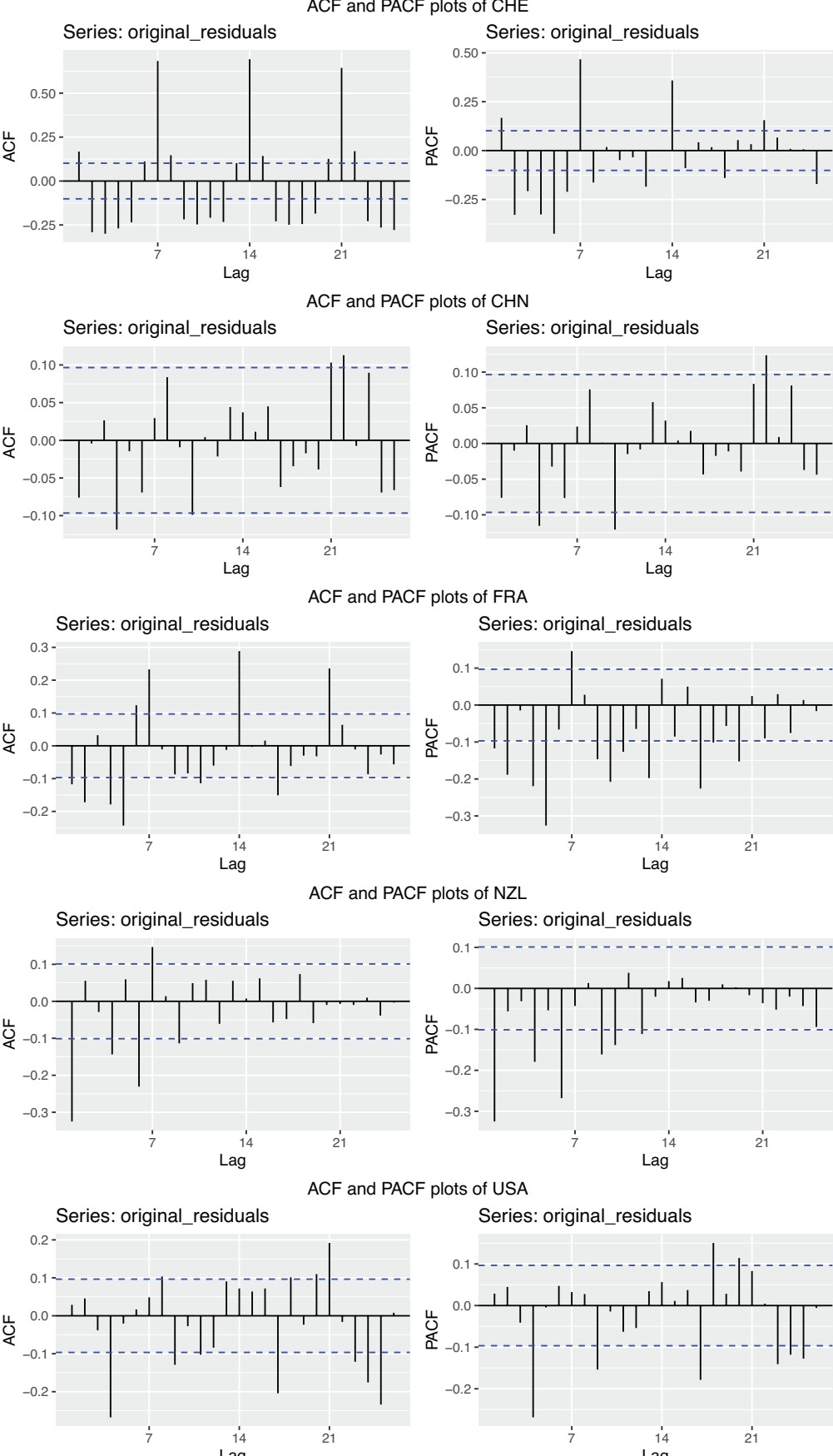

**Appendix 5—figure 2.** Autocorrelation function (ACF) and partial autocorrelation function (PACF) plots of the observations from five different countries. In each row, the two plots are the ACF and PACF plots of the observations from Switzerland (CHE), China (CHN), France (FRA), New Zealand (NZL) and the United States (USA), respectively.

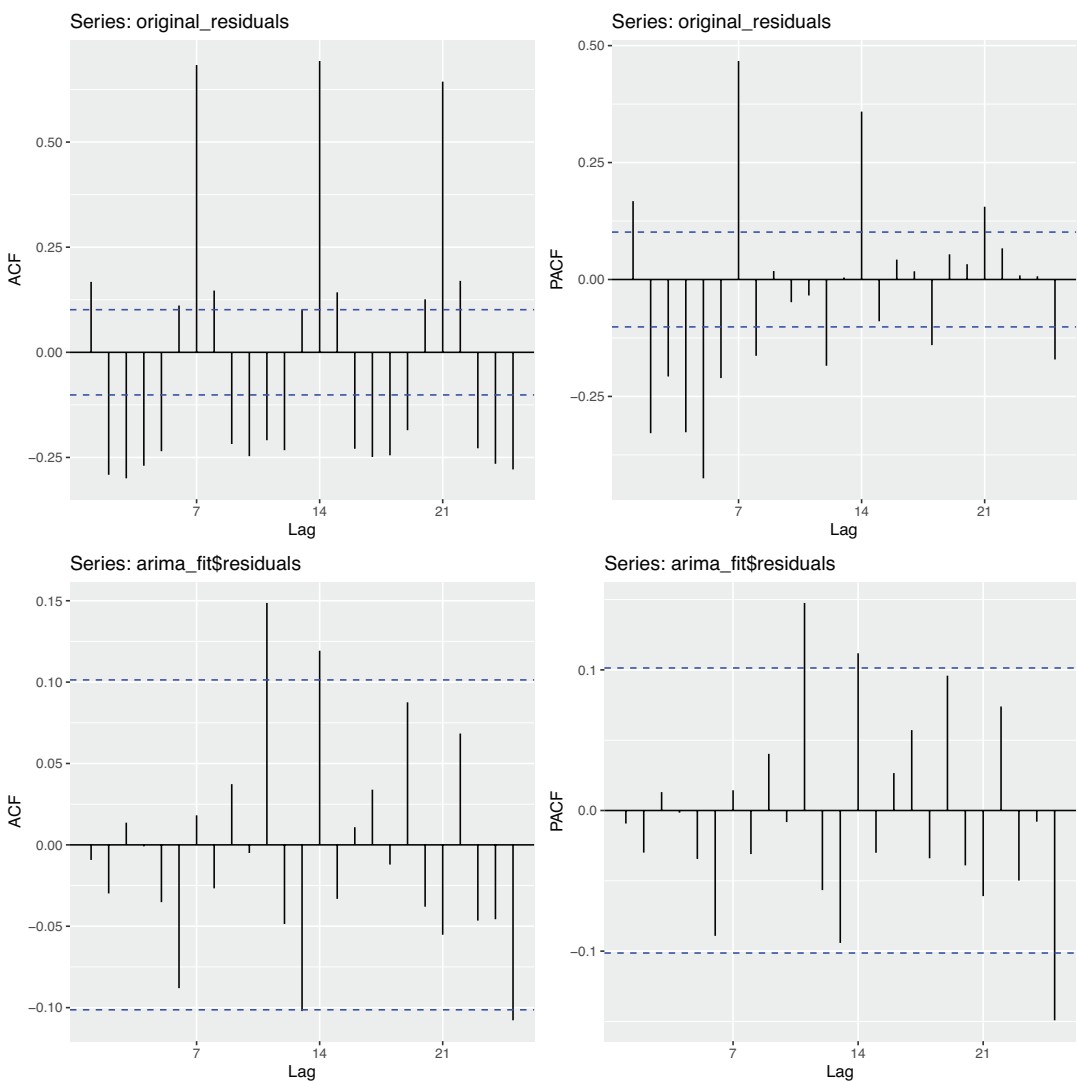

**Appendix 5—figure 3.** Autocorrelation function (ACF) and partial autocorrelation function (PACF) plots of observations from Switzerland and the fitted ARIMA model. The two plots on the upper row are the ACF and PACF plots of the observations from Switzerland. The two plots on the lower row are the ACF and PACF plots of the residuals of the fitted ARIMA model.

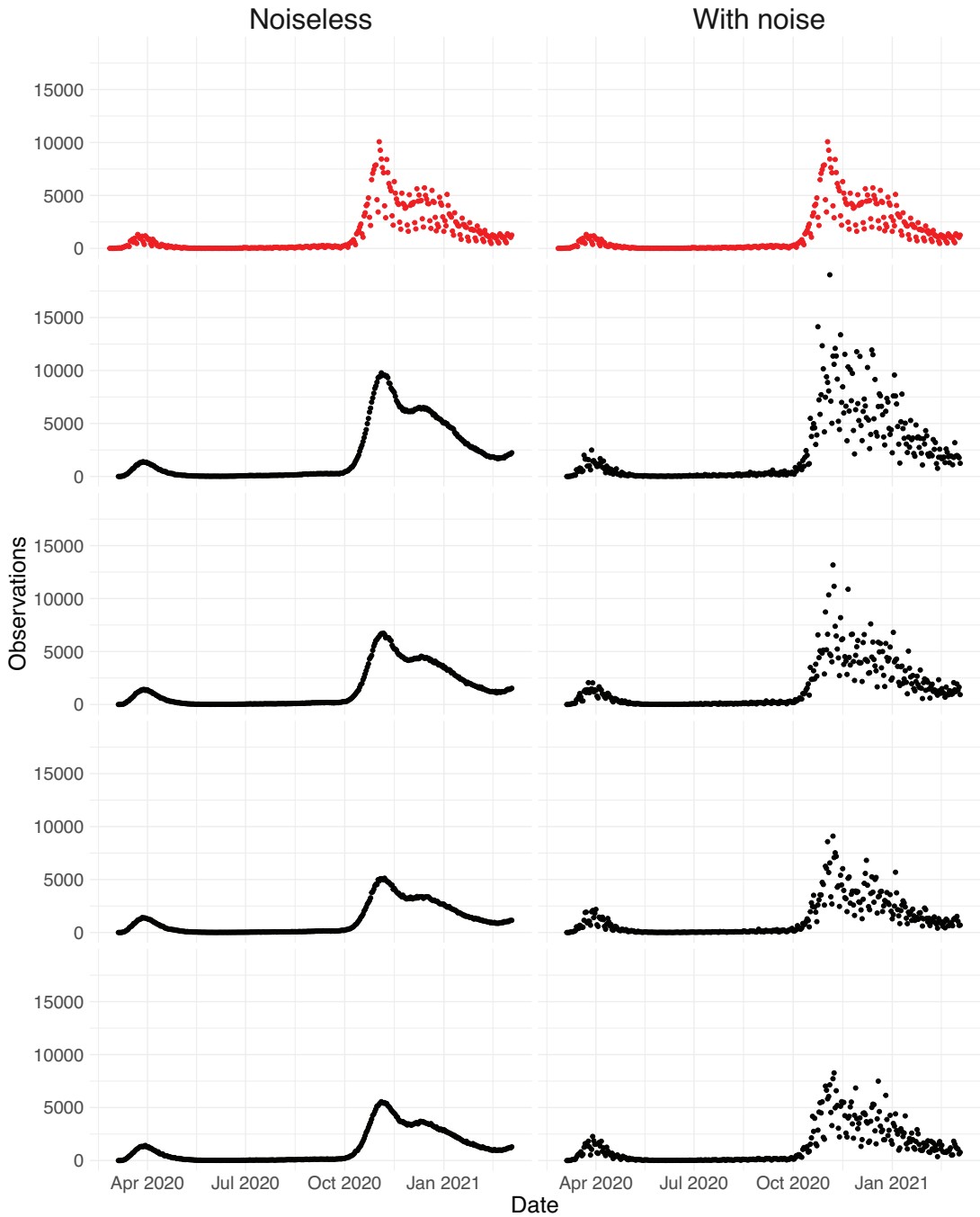

**Appendix 5—figure 4.** Simulated observations with and without noise. The upper row shows the real observations from Switzerland (twice the same). The other four rows show simulated observations, the left column shows simulations without the noise term ($\tilde{D}_t$ in Section 4.7), and the right the simulated observations with the noise term ($D_t$ in Section 4.7).

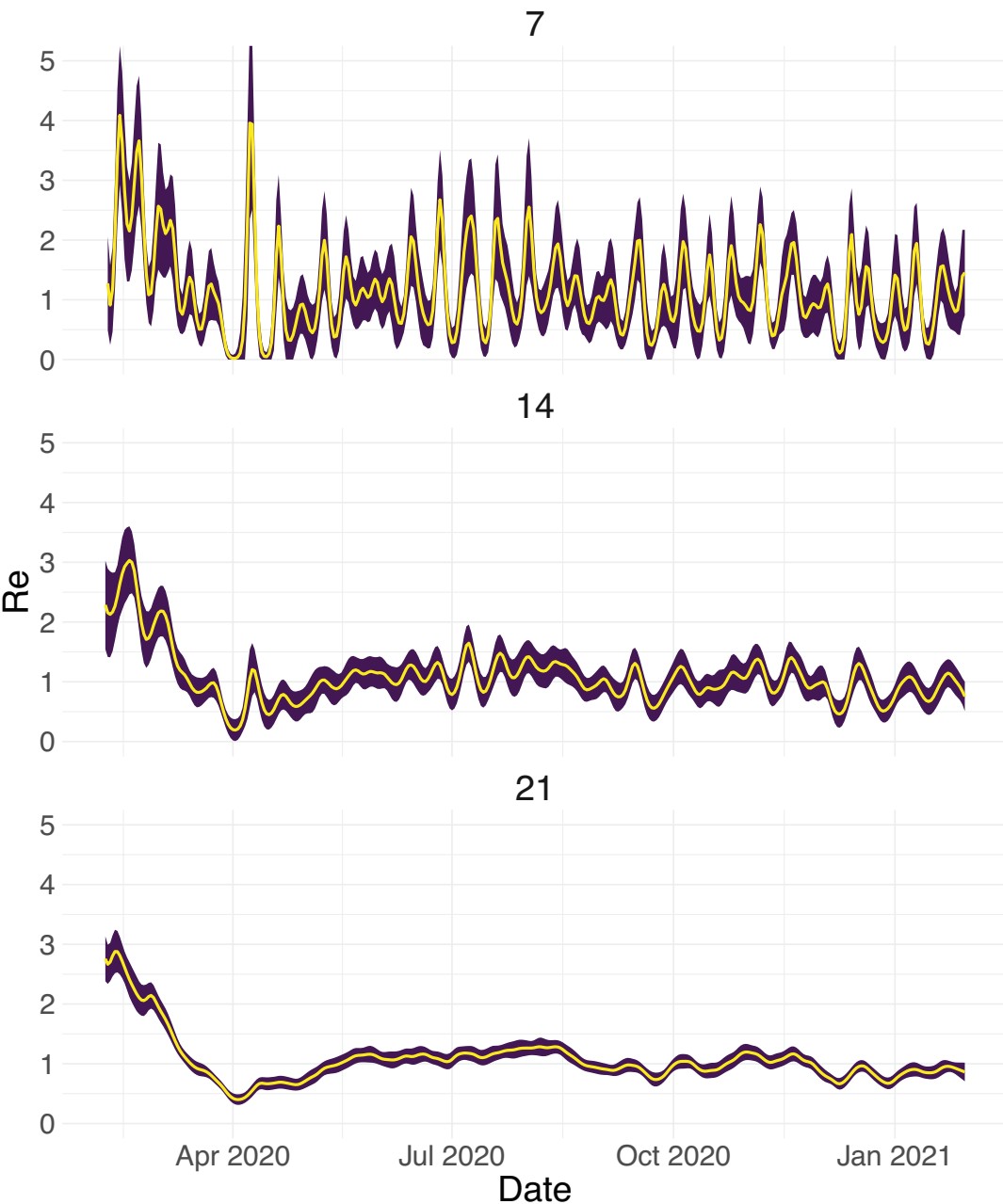

**Appendix 5—figure 5.** Estimated $R_e$ with different smoothing windows. For each trajectory the point estimate is shown with a yellow line, and the purple ribbon indicates the 95% confidence interval.

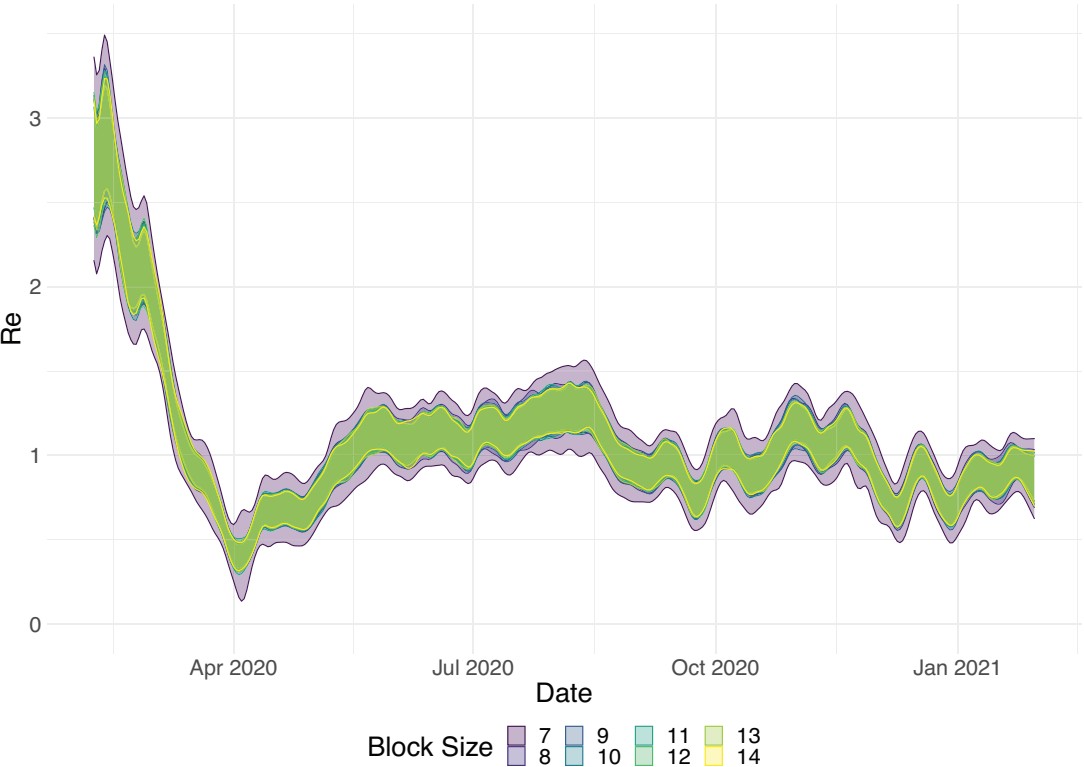

**Appendix 5—figure 6.** Estimated $R_e$ with different block sizes. The ribbons indicate the 95% confidence interval.

## Appendix 6

### SI 50 analysis

Reference date: first day the stringency index exceeded 50 ($SI > 50$).

The 42 included countries: Algeria, Andorra(*), Australia, Austria, Belgium, Canada, Chile, Croatia, Czech Republic, Denmark, Egypt, Estonia, Finland, France, Germany, Greece, Iceland, Indonesia, Iran, Ireland, Israel, Japan, South Korea(*), Lebanon, Malaysia, Mexico, Netherlands, Norway, Philippines, Poland, Portugal, Saudi Arabia, Singapore, Slovenia, Spain, Switzerland, Tajikistan(*), Thailand, United Arab Emirates, United Kingdom, United States, Vietnam.

A star indicates the country was not included in the $\Delta SI$ analysis (e.g. because the biggest jump in SI took place prior to the first possible $R_e$ estimate).

For 37/42 countries the $R_e$ estimate was above one prior to the reference date, and significantly so for 35/42. The countries that reached $R_e < 1$ prior to the reference date were Andorra (17 days prior), Australia (2 day prior), Denmark (3 days prior), Japan (359 days prior), and Vietnam (3 days prior).

### $\Delta SI$ analysis

Reference date: date of the biggest 7 day increase in the SI.

The 45 included countries: Algeria, Australia, Austria, Belarus(*), Belgium, Canada, Chile, Colombia(*), Croatia, Czech Republic, Denmark, Egypt, Estonia, Finland, France, Germany, Greece, Iceland, Indonesia, Iran, Ireland, Israel, Japan, Lebanon, Malaysia, Mexico, Netherlands, New Zealand(*), Norway, Philippines, Poland, Portugal, Russia(*), Saudi Arabia, Serbia(*), Singapore, Slovenia, Spain, Switzerland, Thailand, Turkey(*), United Arab Emirates, United Kingdom, United States, Vietnam.

A star indicates the country was not included in the $SI50$ analysis (e.g. because $SI = 50$ was never reached).

For 41/45 countries the $R_e$ estimate was above one prior to the reference date, and significantly so for 38/45. The countries that reached $R_e < 1$ prior to the reference date were Australia (3 days prior), Denmark (4 days prior), Germany (4 days prior), and Vietnam (10 days prior).

**Appendix 6—table 1.** Investigating the relation between the date of 'lockdown' and the date that the estimated $R_e$ dropped below 1.

The first three columns contain the same information as the first four columns of the main text *Table 1*. The last two columns are analogous to the third ('$\hat{R}_e < 1$ based on confirmed cases') but are based on $R_e$ estimates for hospitalisations and deaths respectively. For each of these observation types, we used our method to determine when the $R_e$ estimate first dropped below 1, and for which dates the corresponding 95% confidence interval contained 1. Further, we used news reports to determine when a country implemented stay-at-home orders (a 'lockdown'). Based on our $R_e$ estimates for confirmed cases, Denmark, Germany, the Netherlands, and Slovenia had 95% confidence intervals that included or were below one before a nationwide lockdown was implemented. For $R_e$ estimates based on COVID-19 deaths, there are also four: Denmark, the Netherlands, Poland, and the United Kingdom. See Appendix 2 for smoothing related caveats.

| Country | Lockdown | $\hat{R}_e < 1$ based on Confirmed cases | $\hat{R}_e < 1$ based on Deaths | $\hat{R}_e < 1$ based on Hospitalisations |
|---|---|---|---|---|
| Austria | 16–03 | 20–03 [20-03, 20-03] | | |
| Belgium | 18–03 | 30–03 [25-03, 03-04] | 26–03 [24-03, 26-03] | 25–03 [24-03, 25-03] |
| Denmark | 18–03 | 10–03 [10-03, 20-06] | 22–03 [18-03, 07-01] | |
| Finland | 16–03 | 02–04 [29-03, 30-04] | 07–04 [25-03, 11-04] | |
| France | 17–03 | 27–03 [23-03, 07-04] | 24–03 [22-03, 26-03] | 27–03 [25-03, 26-03] |
| Germany | 22–03 | 18–03 [17-03, 19-03] | 31–03 [23-03, 04-04] | |
| Ireland | 27–03 | 08–04 [04–04, 15–04] | 05–04 [31-03, 09-04] | 06–04 [06–04, 26–04] |

*Appendix 6—table 1 Continued on next page*

*Appendix 6—table 1 Continued*

| Country | Lockdown | $\hat{R}_e < 1$ based on Confirmed cases | $\hat{R}_e < 1$ based on Deaths | $\hat{R}_e < 1$ based on Hospitalisations |
|---|---|---|---|---|
| Italy | 10–03 | 18–03 [17-03, 19-03] | 14–03 [01–03, 29–05] | |
| Netherlands | 23–03 | 05–04 [22-03, 10-04] | 22–03 [19-03, 02-04] | 26–03 [24-03, 26-03] |
| Norway | 14–03 | 21–03 [17-03, 24-03] | 25–03 [18-03, 08-04] | |
| Poland | 25–03 | 02–04 [31-03, 17-04] | 09–04 [24-03, 01-12] | |
| Portugal | 16–03 | 28–03 [23-03, 15-04] | 28–03 [21-03, 12-04] | |
| Romania | 24–03 | 06–04 [31-03, 29-04] | 17–04 [25-03, 28-04] | |
| Russian Federation | 30–03 | 04–05 [01–05, 08–05] | 18–05 [14-05, 12-12] | |
| Slovenia | 20–03 | 23–03 [13-03, 26-03] | 26–03 [20–03, ≥03-05-2021] | |
| Spain | 14–03 | 26–03 [25-03, 26-03] | | |
| Sweden | | 01–04 [06–03,≥03-05-2021] | 05–04 [13–03,≥03-05-2021] | |
| Switzerland | 17–03 | 22–03 [20-03, 22-03] | 21–03 [18-03, 23-03] | 18–03 [16-03, 18-03] |
| Turkey | 21–03 | 08–04 [01–04, 13–04] | 04–04 [31-03, 06-04] | |
| United Kingdom | 24–03 | 30–03 [28-03, 20-04] | 25–03 [24-03, 25-03] | 29–03 [27-03, 29-03] |

**Appendix 6—table 2.** News and public resources used to determine when a country implemented the first non-pharmaceutical interventions, and a nationwide lockdown.

| Country | First Measure | Lockdown | URL |
|---|---|---|---|
| Austria | 10–03 | 16–03 | https://mrc-ide.github.io/covid19estimates/#/interventions |
| Belgium | 10–03 | 18–03 | https://mrc-ide.github.io/covid19estimates/#/interventions |
| Denmark | 12–03 | 18–03 | https://mrc-ide.github.io/covid19estimates/#/interventions |
| Finland | 16–03 | 16–03 | https://en.wikipedia.org/wiki/COVID-19_pandemic_in_Finland#Response_by_sector |
| France | 29–02 | 17–03 | https://www.politico.eu/article/europes-coronavirus-lockdown-measures-compared/ |
| Germany | 06–03 | 22–03 | https://www.bundesregierung.de/breg-de/themen/coronavirus/besprechung-der-bundeskanzlerin-mit-den-regierungschefinnen-und-regierungschefs-der-laender-1733248 |
| Ireland | 12–03 | 27–03 | https://en.wikipedia.org/wiki/COVID-19_pandemic_in_the_Republic_of_Ireland#Containment_phase |
| Italy | 22–02 | 10–03 | https://www.politico.eu/article/europes-coronavirus-lockdown-measures-compared/ |
| Netherlands | 10–03 | 23–03 | https://www.volkskrant.nl/nieuws-achtergrond/bijeenkomsten-tot-1-juni-verboden-burgemeesters-mogen-handhaven-met-forse-boetes~b41b8508/ |
| Norway | 12–03 | 14–03 | https://www.euractiv.com/section/coronavirus/short_news/norway-update-covid-19/ |
| Poland | 09–03 | 25–03 | https://www.politico.eu/article/europes-coronavirus-lockdown-measures-compared/ |
| Portugal | 11–03 | 16–03 | https://www.politico.eu/article/europes-coronavirus-lockdown-measures-compared/ |
| Romania | 21–02 | 24–03 | https://en.wikipedia.org/wiki/COVID-19_pandemic_in_Romania |
| Russian Federation | 25–03 | 30–03 | https://en.wikipedia.org/wiki/COVID-19_pandemic_in_Russia |
| Slovenia | 09–03 | 20–03 | https://en.wikipedia.org/wiki/COVID-19_pandemic_in_Slovenia |
| Spain | 10–03 | 14–03 | https://www.elmundo.es/espana/2020/03/13/5e6b844e21efa0dd258b45a5.html |
| Sweden | 11–03 | | |
| Switzerland | 28–02 | 17–03 | 'Verordnung 2 über Massnahmen zur Bekämpfung des Coronavirus (COVID-19)' |
| Turkey | 12–03 | 21–03 | https://en.wikipedia.org/wiki/COVID-19_pandemic_in_Turkey#Government_response |
| United Kingdom | 12–03 | 24–03 | https://www.bbc.com/news/uk-52012432 |

