## [Editor Report]

Understanding the trajectory of epidemic growth and predicting it in real-time is an important goal of epidemiological modelling. This work aggregates data from 170 countries in an effort to better understand how the effective reproduction number of SARS-CoV-2 spread evolved over time and across the world.

---

## [Decision Letter]

**Decision letter after peer review:**

Thank you for submitting your article "Estimation and worldwide monitoring of the effective reproductive number of SARS-CoV-2" for consideration by *eLife*. Your article has been reviewed by 2 peer reviewers, and the evaluation has been overseen by a Reviewing Editor and Miles Davenport as the Senior Editor. The reviewers have opted to remain anonymous.

Essential revisions:

1) In addition to the stringency index, please incorporate mobility in selected countries where relevant data are available.

2) Moreover, please integrate additional data types, eg case series, hospitalizations and deaths, to estimate a single set of R_e (t), rather than separate analyses on each data stream.

3) Please address the uncertainty estimation procedure as queried by Reviewer #1.

Reviewer #1 (Recommendations for the authors):

1) Explain more for the simulation approach in the paper, such as why the noise should be computed in this way, instead of just based on the theoretical delay distribution from infection to death/hospitalization/reported, to simulate the input time series.

2) More simulations to validate the real-time property of the estimates of Rt

3) A potential framework that can use both time series for cases, hospitalization and death to jointly infer the infection time series.

The results are easy to be assessed and presented clearly, which is good. I am not sure how I can get the data and also the code to replicate the analysis. Ideally it should be available.

(Note from Ed: Data and code must be available to conform with *eLife* data policies)

---

## [Author Response]

Essential revisions:1) In addition to the stringency index, please incorporate mobility in selected countries where relevant data are available.

We have now integrated mobility data into our dashboard, similar to our stringency index and vaccination time series. We show both Apple and Google mobility data. All these traces can be downloaded and compared against our Re estimates. We have also incorporated this data into the empirical analysis of Figure 5 in the manuscript.

2) Moreover, please integrate additional data types, eg case series, hospitalizations and deaths, to estimate a single set of R_e (t), rather than separate analyses on each data stream.

We think it is a strength of our approach to show the different traces in parallel, as this allows direct comparison during periods where one trace may be more or less credible. We have revised the text to make clear early on that we estimate separate Re values, and what the advantages of this approach are.

3) Please address the uncertainty estimation procedure as queried by Reviewer #1.

We believe there was some misunderstanding here regarding the logic behind our simulation set-up. We have revised the corresponding method sections, specifically adding more information on why and how we incorporate observation noise into our simulations.

Reviewer #1 (Recommendations for the authors):1) Explain more for the simulation approach in the paper, such as why the noise should be computed in this way, instead of just based on the theoretical delay distribution from infection to death/hospitalization/reported, to simulate the input time series.

We added more explanation about our simulation approach to the corresponding method section (lines 488-518). We found that simulated observations based only on the delay from infection to reporting did not show the same amount of variability as real case data from around the world. We added reporting noise to the simulations to be able to accurately assess the coverage of our method.

Please also refer to our answer to the public review above, for a more detailed overview of our simulation set-up.

2) More simulations to validate the real-time property of the estimates of Rt

As suggested, we performed an additional simulation study to investigate the accuracy and stability of the last possible Re estimate. We present this analysis in a new results paragraph (subsection "Stability of Re estimates in an outbreak monitoring context"; line 121) and Appendix 4 – Figure 1. This analysis emphasizes the need to be cautious with the most recent Re estimates when these values are directly policy relevant.

We revised the wording of our manuscript everywhere to replace the ambiguous "near real-time" by "timely", and added text to highlight the delay between latest Re estimates and latest case observations.

3) A potential framework that can use both time series for cases, hospitalization and death to jointly infer the infection time series.

Please refer to our response to item #1 of the public review (for reviewer 1).

The results are easy to be assessed and presented clearly, which is good. I am not sure how I can get the data and also the code to replicate the analysis. Ideally it should be available.(Note from Ed: Data and code must be available to conform with eLife data policies)

The information on data sources is provided in the Data subsection (4.6) of the Materials and methods section, with additional information in Supplementary File S1.

The information on software repositories is provided at the very end of the main text of the manuscript, in the Materials and methods section (subsection 4.8).

Specifically, the source code of the software pipeline is openly accessible at https://github.com/covid-19-Re/shiny-dailyRe, and the code necessary to reproduce the figures and analyses presented in this paper is available at https://github.com/covid-19-Re/paper-code.